



# Modeling of the in situ state of stress in elastic layered rock subject to stress and strain-driven tectonic forces

Vincent Roche *, Mirko van der Baan,

1: Dept. of Physics, CCIS, University of Alberta, Edmonton, AB, T6G 2E1, Canada
*Correspondence to*: V. Roche (roche@ualberta.ca)

**Abstract:** In this study we describe and compare eight different strategies to predict the depth variation of stress within a layered rock formation. This reveals the inherent uncertainties in stress prediction from elastic properties and stress measurements, as well as the geologic implications of the different models. The predictive strategies are based on well log data and in some cases on in situ stress measurements, combined with the weight of the overburden rock, the pore pressure, the depth variation in rock properties, and tectonic effects. We contrast and compare stresses predicted purely using theoretical models with those constrained by in situ measurements. We also explore the role of the applied boundary conditions mimicking two fundamental models of tectonic effects, namely the stress or strain-driven models. In both models layer to layer tectonic stress variations are added to initial predictions due to vertical variation in rock elasticity, consistent with natural observations, yet describing very different controlling mechanisms. Layer to layer stress variations are caused by either local elastic strain accommodation for the strain-driven model, or stress transfers for the stress-driven model. As a consequence, stress predictions can depend strongly on the implemented prediction philosophy and the underlying implicit and explicit assumptions, even for media with identical elastic parameters and stress measurements. This implies that stress predictions have large uncertainties, even if local measurements and boundary conditions are honored.

## 1. Introduction

Knowledge of in situ stress magnitudes and their spatial variability is critical to understand the upper crust stress and strain [Schmitt et al., 2012; Reiter et al., 2014; Reiter and Heidbach, 2014] which in turn has strong implications for seismotectonics (e.g., earthquake magnitudes [Davies et al., 2012; Langenbruch and Shapiro 2014; Busetti and Reches, 2014; Scholz, 2015] and their locations [Sibson, 1982; Zoback and Gorelick, 2012; Zakharova and Goldberg, 2014]), structural geology (e.g., fault behavior and slip tendency [Gross et al., 1997; Gudmundsson, 2011]), volcanotectonics (e.g. prediction of dike paths and eruption forecasting [Gudmundsson, 2006]). It is also key for the civil engineering, mining and energy industries covering topics as diverse as hydraulic fracturing, rock stability, fluid circulation [Simonson et al., 1978; Van Eekelen, 1982; Warpinski et al., 1985; Hopkins et al., 1997; King, 2010; Fisher and Warpinski, 2011; Davies et al., 2012], and for determining the likelihood of felt seismicity due to human activities, for instance, because of hydraulic fracturing or salt-water disposal [Frohlich, 2012; Ellsworth, 2013; Weingarten et al., 2015; Atkinson et al., 2016; Van der Baan and Calixto, 2016].

In situ stresses may be assessed using different techniques, such as extended leak-off tests, hydraulic fracturing treatments, borehole breakouts, earthquake source mechanisms, geological indicators, etc. [Terzaghi, 1962; Hast 1967;



Zoback et al., 1985; Amadei and Stephansson, 1997; Zang and Stephansson, 2010; Zoback, 2010; Schmitt et al., 2012; Reiter et al., 2014; Reiter and Heidbach, 2014]. In rocks composed of layers with different elastic properties like in sedimentary or volcanic areas, layer to layer variations in horizontal stresses may arise. This phenomenon occurs in

several lithology types such as coal, mudstone, siltstone, sandstone, limestone, shales, lava flows, intrusions, and pyroclastics [Haimson and Rummel, 1982; Warpinski et al., 1985; McLellan, 1987; Evans et al., 1989; Warpinski and Teufel, 1991; Cornet and Burlet, 1992; Gunzburger and Cornet, 2007; Cornet and Röckel, 2012]. Such stress variations are well documented, but local information on the stress field is often sparse and incomplete; a continuously sampled stress profile is rarely available, because extended leak off tests often concentrate only on the target formations and are

rarely published. In this case models have to be used to assess the stress variations. We focus on the 1D depth variation in stress, which involves models based on analytic formulations [McGarr, 1988; Gunzburger and Cornet, 2007; Jaeger et al., 2009] and numerical modeling [Teufel and Clark, 1984, Gudmundsson, 2006; Roche and Van der Baan, 2015].

The simplest model, called the lithostatic model, assumes an isotropic state of stress equal to the vertical stress as calculated from the weight of the overburden rock [Jaeger et al., 2009]. Alternatively, in the critical state-of-stress

model, stress depends on the frictional strength of pre-existing fractures and faults [Zoback, 2010]. This model defines bounds on the magnitude of the principal stresses [Brace and Kohlstedt, 1980] because even a small stress increase would cause slip on faults, resetting stresses to the critical state. Otherwise, stress prediction can be based on elastic rock behavior. For instance, in the uniaxial strain model, stress depends on the Poisson's ratio [Warpinski et al., 1985; McLellan, 1987; Savage et al., 1992; Addis et al., 1996; Jaeger et al., 2009].

Processes such as tectonic effects, uplift or subsidence [Haxby and Turcotte, 1976; McGarr, 1988; Gunzburger and Cornet, 2007], temperature change [Voight and St. Pierre, 1974; Haxby and Turcotte, 1976; McGarr, 1988; Blanton and Olson, 1999], viscous behavior [Gunzburger and Cornet, 2007; Cornet and Röckel, 2012], can be taken into account by applying stress updates to initial stress models. This is a common methodology to predict stresses in the crust beyond individual measurements.

An alternative prediction strategy is to assume that the state of stress in the crust is close to the maximum strength that rocks can support at the large scale (i.e. scales of a kilometer or more). The critical stress model then becomes applicable. Or, this model can provide bounds for the minimum and maximum principal stresses [Brace and Kohlstedt, 1980; Townend and Zoback, 2000; Zoback, 2010; Konstantinovskaya at al., 2012; Meixner et al., 2014] since the Earth may not be critically stressed everywhere. The rationale for this approach is that faults and fractures are the weakest

structures in rocks. Therefore the stress differentials cannot exceed their shear strength. Any excess stress will lead to rupture, resetting the stresses to below the critical stresses.

The use of the critical model to obtain bounds for the upper and lower stresses was advocated by Brace and Kohlstedt (1980) since stress predictions from the elastic properties of rock is prone to large uncertainty, for areas where the crust has a long and complex tectonic history, possibly resulting in non-elastic behavior. Yet, stress predictions based

on elasticity are commonly used, notably because elastic properties are easier to obtain from well logs than strength properties and in situ stress measurements. In addition, numerous applications in seismotectonics, volcanotectonics, structural geology and hydrocarbon exploitation require more detailed stress information than upper and lower bounds.

The objective of this work is to analyse different methodologies for 1D stress prediction. We explore notably the role of boundary conditions in the models, i.e. whether the tectonic forces are strain or stress-driven. Then, we assess the

inherent uncertainty in stress predictions due to lack of information and/or the use of different updating strategies. Several models are used and combined, focusing on two processes: the additive tectonic forces and the effect of depth variations



in the elastic parameters. For illustration, all modeling strategies are applied to a field study case, where the resulting stress predictions are compared and contrasted.

**2. Stress prediction strategies**

**2.1. Tectonic driving forces and stress corrections**

For models depending on elastic rock behavior, prediction of the stress state is based on an initial stress model that is then modified by adding stress corrections (increments or decrements) to the horizontal stress components, that is

$$\sigma'_{hl} = \sigma_{hli} - P_p + \Delta\sigma_{hl} \text{ and } \sigma'_{Hl} = \sigma_{Hli} - P_p + \Delta\sigma_{Hl},$$ (1)

where $\sigma'_{hl}$ and $\sigma'_{Hl}$ are the predictions for respectively the minimum and maximum, effective, horizontal stress, $\sigma_{hli}$ and
$\sigma_{Hli}$ are the local initial stress predictions, and $\Delta\sigma_{hl}$ and $\Delta\sigma_{Hl}$ are the local tectonic stress corrections. The initial stress predictions $\sigma_{hli}$ and $\sigma_{Hli}$ can originate from a uniaxial model, Eq. (A.2) in Appendix A [Voight and St. Pierre, 1974; Haxby and Turcotte, 1976; Savage 1992; Blanton and Olson 1999], or a lithostatic model, Eq. (A.1) [McGarr 1988; Roche et al, 2013; Roche and Van der Baan, 2015]. The initial model represents our best guess for the most representative baseline stress state. The stress corrections reflect processes such as erosion, sedimentation, temperature change, tectonic activity.
Various approaches exist to estimate the desired stress corrections $\Delta\sigma_{hl}$ and $\Delta\sigma_{Hl}$. For instance, knowledge of the true magnitudes of the in situ local stresses $\sigma_{hl}$ and $\sigma_{Hl}$, permits their direct computation using

$$\Delta\sigma_{Hl} = \sigma_{Hlm} - \sigma_{Hli} \text{ and } \Delta\sigma_{hl} = \sigma_{hlm} - \sigma_{hli},$$ (2)

where $\sigma_{Hlm}$ and $\sigma_{hlm}$ are local, independent measurements of both horizontal stresses.

Alternatively, one could assume that the magnitude of the tectonic stress corrections $\Delta\sigma_{hl}$ and $\Delta\sigma_{Hl}$ increase until
the initial stresses $\sigma_{hli}$ and $\sigma_{Hli}$ reach the critical stress model in the target formation (see Appendix A for background on the critical stress model). In this case, the stress corrections for the different stress regimes are obtained by combining Eq. (1) and Eq. (A3), yielding

*Normal stress regime:* $\Delta\sigma_{hl} = \dfrac{|\sigma_v - P_p|}{\left(\sqrt{\mu^2+1}+\mu\right)^2} + P_p - \sigma_{hli},$ (3)

*Thrust fault regime:* $\Delta\sigma_{Hl} = \left(\sqrt{\mu^2+1}+\mu\right)^2\left[|\sigma_v - P_p|\right] + P_p - \sigma_{Hli},$ (4)

*Strike slip regime:* $\Delta\sigma_{Hl} = \left(\sqrt{\mu^2+1}+\mu\right)^2\left[|\sigma_{hli} - P_p + \Delta\sigma_{hl}|\right] + P_p - \sigma_{Hli};$

$$\Delta\sigma_{hl} = \dfrac{|\sigma_{Hli} - P_p + \Delta\sigma_{Hl}|}{\left(\sqrt{\mu^2+1}+\mu\right)^2} + P_p - \sigma_{hli}.$$ (5)

The local critical-state stress corrections $\Delta\sigma_{hl}$ or $\Delta\sigma_{Hl}$ for the normal and thrust fault regimes, Eq. (3) and Eq. (4), respectively, may be obtained without any measurements. The local stress corrections applied on the intermediate principal stress, $\Delta\sigma_{hl}$ or $\Delta\sigma_{Hl}$ for the thrust and normal fault regimes, respectively, cannot be calculated in this case because



the critical stress is based on the magnitude of the maximum and minimum principal stresses. For the strike-slip regime both stress corrections $\Delta\sigma_{hl}$ and $\Delta\sigma_{Hl}$ are interdependent in Eq. (5). They can be calculated only if one is known from measurements using Eq. (2).

It is important to distinguish here between the regional stress or strain corrections (boundary conditions) and the resulting local stress and strain variations (internal conditions). The local stress corrections, calculated with Eq. (2), (3),
(4), or (5), are defined for a specific layer. It is not possible to apply them to all other layers because in layered rocks stresses are prone to change as a function of depth since the stress corrections depend on the elastic rock properties. To calculate the local stress corrections for each layer, one needs to find the regional (external) tectonic perturbations applied to the rock formation that lead to the actual in situ local stresses in all layers. For convenience we will refer to the regional stress or strain perturbations as the regional perturbations, and the resulting local stress changes as the local stress
corrections.

The regional perturbations may correspond either to a regional strain or stress perturbation, referred to as the strain and stress-driven models, respectively. In the first case horizontal strain results from a displacement caused by plate tectonics [Savage 1992; Blanton and Olson 1999; Beaudoin et al., 2011; Song and Hareland 2012, Reiter and Heidbach, 2014]. In the second case, regional tectonic forces impose external stress instead of strain [Teufel and Clark 1984; Bourne
2003; Roche et al., 2013]. These two fundamental models are detailed in the next sub-sections.

## 2.2. Strain-driven model

For the strain-driven model a biaxial strain model is appropriate in order to provide the magnitude of the local stress corrections since no variation in the overburden stress is assumed [Jaeger et al., 2009]. In an isotropic elastic rock, the local stress corrections $\Delta\sigma_{hl}$ and $\Delta\sigma_{Hl}$ are then linked to the tectonic regional strain perturbations, $\Delta\varepsilon_{hr}$ and $\Delta\varepsilon_{Hr}$, by
[Savage, 1992; Economides and Nolte, 2000]:

$$\Delta\sigma_{hl} = \frac{E}{1-\upsilon^2}\Delta\varepsilon_{hr} + \frac{E\upsilon}{1-\upsilon^2}\Delta\varepsilon_{Hr}, \; \Delta\sigma_{Hl} = \frac{E}{1-\upsilon^2}\Delta\varepsilon_{Hr} + \frac{E\upsilon}{1-\upsilon^2}\Delta\varepsilon_{hr}, \tag{6}$$

and reciprocally [Jaeger et al., 2009]

$$\Delta\varepsilon_{Hr} = \frac{\left(1-\upsilon^2\right)}{E}\Delta\sigma_{Hl} - \frac{\upsilon(1+\upsilon)}{E}\Delta\sigma_{hl}, \; \Delta\varepsilon_{hr} = \frac{\left(1-\upsilon^2\right)}{E}\Delta\sigma_{hl} - \frac{\upsilon(1+\upsilon)}{E}\Delta\sigma_{Hl}. \tag{7}$$

The local stress corrections $\Delta\sigma_{hl}$ and $\Delta\sigma_{Hl}$ depend on the vertical variation in Young's modulus $E$ and Poisson's
ratio $\upsilon$. Horizontal stress corrections increase with increasing Young's modulus for regional shortening (Figure 1A) and decrease for regional lengthening (Figure 1B). Both horizontal regional strain perturbations, $\Delta\varepsilon_{Hr}$, $\Delta\varepsilon_{hr}$, are needed in order to assess the vertical variation in horizontal stresses.

Assuming an average Poisson's ratio of 0.3 or less, the stress correction in the perpendicular direction is merely 30% of the one in the direction of the strain perturbation. Consequently, in cases where the regional strain correction
occurs mainly in one direction, the second term in Eq. (6) and Eq. (7) is sometimes neglected, producing

$$\Delta\sigma_{hl} = \frac{E}{1-\upsilon^2}\Delta\varepsilon_{hr} \text{ and } \Delta\sigma_{Hl} = \frac{\upsilon E}{1-\upsilon^2}\Delta\varepsilon_{hr} \text{ if } \Delta\varepsilon_{Hr} \approx 0, \tag{8}$$





$$\Delta\sigma_{Hl} = \frac{E}{1-\upsilon^2}\Delta\varepsilon_{Hr} \text{ and } \Delta\sigma_{hl} = \frac{\upsilon E}{1-\upsilon^2}\Delta\varepsilon_{hr} \text{ if } \Delta\varepsilon_{hr} \approx 0, \tag{9}$$

For instance, Eq. (8) may be used for a normal stress regime where regional strain corresponds mainly to a lengthening in one horizontal direction. Equation 9 fits better a thrust fault regime with a single direction of shortening. The regional strain perturbations, $\Delta\varepsilon_{hr}$, or $\Delta\varepsilon_{Hr}$, may be locally derived from the horizontal stress corrections $\Delta\sigma_{hl}$, or $\Delta\sigma_{Hl}$, respectively (Eq. (2)), as obtained from in situ stress measurements. The problem becomes more complex if the second lateral regional strain perturbation is not negligible.

Figures 1A and 1B illustrate conceptually the behavior of the strain-driven model. For such a model, the local strain corrections $\Delta\varepsilon_{Hc}$ and $\Delta\varepsilon_{Hs}$, or $\Delta\varepsilon_c$ and $\Delta\varepsilon_s$, are equal to the regional strain perturbations, $\Delta\varepsilon_{Hr}$ and $\Delta\varepsilon_{hr}$ in each layer, when a bilayer elastic medium is subjected to a compressive stress regime (a shortening in the direction of the maximum principal stress, i.e., positive $\Delta\varepsilon_{hr}$ in figure 1A) or a normal stress regime (a lengthening in the direction of the minimum principal stress, i.e., negative $\Delta\varepsilon_{Hr}$ in figure 1B), since the two parts of the moving wall are vertically aligned.

Conversely, the local stress corrections $\Delta\sigma_{Hl}$ and $\Delta\sigma_{hl}$ change in each layer to account for uniform local strain perturbations due to different Young's moduli and Poisson's ratios. Consequently, the final stress profile shows layer to layer stress variations predominantly in the direction of the regional strain perturbations (i.e., $\sigma_H$ and $\sigma_h$ in figure 1A and 1B, respectively), and to a lesser extent in the orthogonal direction (i.e., $\sigma_h$ and $\sigma_H$ in figure 1A and 1B, respectively).

Figure 2A shows the final local stresses, $\sigma_3$, and the local stress corrections, $\Delta\sigma_{hl}$, calculated analytically with Eq. (8) and Eq. (9), in various bi-layer sections subjected to a regional shortening The increase in lithostatic stress with depth due to the weight of the overburden rock has been ignored in order to highlight better the effect of the layering. The local stress corrections $\Delta\sigma_{hl}$ and predicted stresses $\sigma_3$ depend solely on the local Young's moduli and Poisson's ratios of the individual layers; they are independent of the rock behavior or properties of the surrounding layers, since no stress transfer or interaction occurs between layers. For instance predicted stresses $\sigma_3$ are equal in the layers with a Young's modulus of 50 GPa, whether it is compliant or stiff compared with surrounding layers; predicted stress corrections are independent of the stiffness contrasts between layers.

## 2.3. Non-coupled stress-driven model

In the stress-driven model, the tectonic perturbations are driven by constant tectonic stresses, $\Delta\sigma_{hr}$ and $\Delta\sigma_{Hr}$, applied to the rocks [Teufel and Clark, 1984; Mandl, 2000]. In layered rocks, each layer then deforms independently if the layers are not coupled. The regional tectonic stresses induce local strain perturbations. For instance, the local strain correction for the non-coupled layers $\Delta\varepsilon_{hnc}$ in the direction of the minimal principal stress can be calculated using

$$\Delta\varepsilon_{hnc} = \frac{\left(1-\upsilon^2\right)}{E}\Delta\sigma_{hr} - \frac{\upsilon\left(1+\upsilon\right)}{E}\Delta\sigma_{Hr}, \tag{10}$$

following Eq. (7). The local strain corrections are discontinuous across layer boundaries due to the variation of the elastic properties and slip may occur at the interfaces between layers. The local stress corrections, $\Delta\sigma_{hl}$ and $\Delta\sigma_{Hl}$, are constant and equal to the regional stress perturbations, $\Delta\sigma_{hr}$ and $\Delta\sigma_{Hr}$, in each layer, creating a homogeneous stress profile. This model is illustrated in figures 1C and 1D, in which the bilayer elastic medium from Figures 1A,B is subjected to a regional stress that is either positive (compressional) in the direction of the maximum principal stress (i.e., $\Delta\sigma_{hr}$ in figure 1C), or negative (extensional) in the direction of the minimum principal stress (i.e., $\Delta\sigma_{Hr}$ in figure 1D). In both cases, the local





stress and stress corrections $\sigma_h$ and $\Delta\sigma_{hl}$ in figure 1C, and $\sigma_H$ and $\Delta\sigma_{Hl}$ in figure 1D are constant and the resulting final stress profiles are independent of depth and the actual rock properties.

**2.4. Coupled stress-driven model**

If the layers are coupled together, slip cannot occur at the interface between the layers, and the strain has to be continuous throughout the medium. The regional stresses, $\Delta\sigma_{hr}$ and $\Delta\sigma_{Hr}$, and the regional strain perturbations, $\Delta\varepsilon_{hr}$, or $\Delta\varepsilon_{Hr}$, are constant across layer boundaries. The contrast in elastic properties then produces stress transfer between layers. Local stress corrections for individual layers thus depend on the elastic properties of all layers, contrary to the strain-driven model [Teufel and Clark, 1984; Mandl, 2000; Bourne, 2003]. For a compressive regional stress perturbation $\Delta\sigma_{Hr}$, a

compliant layer is restrained from further shortening by any surrounding stiff layers and the local stress correction $\Delta\sigma_{Hl}$ incorporates an additional layer-parallel extensional stress (Figure 1E). Hence, the local stress $\sigma_H$ decreases in this layer and, in return, it increases in the stiff layer because of an additional layer-parallel compressive stress correction imposed by the shortening of the compliant layer. The opposite situation is encountered for an extensional regional stress perturbation, $\Delta\sigma_{hr}$, due to the elongations of the stiff and compliant layers, requiring different local stresses to overcome

the different Young's moduli (Figure 1F). Notice that the coupled stress-driven model can lead to highly similar local stress profiles as for the strain-driven model (Figures 1A and 1B), although the mechanisms are very different.

Numerical modeling is commonly used to compute the resulting stress profile for more complex media [Teufel and Clark, 1984; Bourne, 2003; Roche et al., 2013; Roche and Van der Baan, 2015]. Appendix B contains the analytic formulation for this model for the simple case of two layers submitted to a tectonic force in a single direction.

Figure 2B and C shows the profiles of the final local stresses, $\sigma_3$, and the local stress corrections, $\Delta\sigma_{hl}$, calculated with numerical modeling and the analytic solutions (Eq. B.11 and Eq. B.12 in Appendix B), for the same media and initial lithostatic stress model as in Figure 2A. The media are submitted to a negative (extensional) 10 MPa regional stress perturbation, $\Delta\sigma_{hr}$.

For the numerical solution, the local stress corrections $\Delta\sigma_{hl}$ reach extrema at the layer interface and tend to a

constant value at the top and bottom, equal to the initial lithostatic stress minus the 10 MPa regional stress correction. The analytic formulation reproduces only the extreme values of the local stress corrections but not the local stress variation within a layer. For periodic media composed of two alternating layers, the local stresses are influenced by both surrounding layers. Consequently, the horizontal stress profile becomes almost constant within a layer [*Roche et al., 2013*], converging to the analytic solutions.

In the coupled stress-driven model, the magnitude of the local stress correction only depends on the contrast in Young's moduli, not their absolute values. The layer to layer local stress variation increases with increasing contrast in Young's moduli. For instance, in figure 2B, the local stress correction and the layer to layer stress variations are nearly constant for a ratio of 2, for different combinations of Young's moduli. They are significantly higher for a ratio of 5, despite similarity in Young's modulus in the different configurations. This mechanism is thus different from the strain-

driven model, where the local stress corrections depend only on the absolute values of the individual Young's moduli, but not their ratios (Figure 2A). The stress transfer effect also depends on the thickness of the individual layers (see Eq. B.8 and Eq. B.9), as confirmed by numerical studies [*Roche et al., 2013*], contrary to the strain-driven model.





### 3. Strategy

Eight different predictive strategies (Table 1) are carried out in order to explore the role of (1) the initial state of stress (uniaxial or lithostatic), (2) the implemented boundary conditions (strain-driven or stress-driven) and (3) the benchmarking and updating procedure (based entirely on the conceptual critical stress model, or available in situ local stress measurements) on final predicted stress profiles. The four steps of calculation for the eight predictive strategies are presented in a synthesis view in Figure 3 and detailed in the next sub-sections.

The first step corresponds to the definition of the initial state of stress used as a base for the stress prediction. The second step is the definition of the reference stresses. The third step incorporates the regional stress perturbations, by combining the initial stresses and the references (stress measurements or critical stress model). In the final step, the initial stresses are updated by including the tectonic effects using either the stress-driven or strain-driven model.

#### 3.1. Step 1: initial stresses

The stress predictions are based on two initial models with both horizontal stresses equal either to the uniaxial or lithostatic stress (Figure 3). The average density is used to calculate the lithostatic stresses $\sigma_l$, Eq. (A.1). We use the depth-dependent Poisson's ratios and pore pressures to calculate the uniaxial stresses $\sigma_u$, Eq. (A.2). We assume a Biot pore-pressure coefficient equal to 1. For simplicity, the pore pressure profile is calculated using a constant pore pressure gradient. By comparing the stress predictions obtained with these two initial models, we explore the possible range of local stress corrections because the stress predictions obtained assuming an initial lithostatic and uniaxial state of stress are likely the upper and lower limits of the tectonic perturbations, respectively.

#### 3.2. Step 2: preparation of the reference stresses

To assess the magnitude of the tectonic effects, we compare the initial stress profile with a reference stress model that is either the critical stress model or locally measured stresses. The choices for initial and reference stresses produce four base predictions. These different choices for the reference stress profiles allow us to compare stress predictions based entirely on conceptual models with those using in situ stress measurements.

In the case of observed data, the local stress corrections $\Delta\sigma_{hl}$ and $\Delta\sigma_{Hl}$ are directly calculated using Eq. (2). For the critical model, we must first assume a specific stress regime. For a normal fault regime and a thrust fault regime, the calculation of the critical horizontal stress is straightforward since the vertical stress becomes the maximum principal stress, or the minimum principal stress, respectively. For the strike slip regime, one needs at least one more data point to calibrate either the minimum, or the maximum principal stress at a specific depth. This could be done with in-situ stress measurements available in one horizontal direction. In this study we assume a normal stress regime. The critical horizontal stress $\sigma_{3c}$ is calculated using, the pore pressure profile, an average friction of 0.6 and the maximum principal critical stress $\sigma_{1c}$ equals the vertical stress $\sigma_v$, Eq. (A.3).

#### 3.3. Step 3: Computation of the tectonic perturbations (boundary conditions)

In order to compute the regional stress or strain perturbation, we compare the initial stress profiles to the reference stress profiles. This comparison could be done in the directions of both horizontal principal stresses, if the relevant reference stresses are available. No stress measurements are available for the maximum horizontal principal stress in the following case study. In order to simplify the problem, avoiding extra assumptions, and to be able to compare stress




predictions obtained with both reference stresses independently, we postulate that only the minimum regional stress $\Delta\sigma_{hr}$
or strain $\Delta\varepsilon_{hr}$ perturbation exists and that the maximum regional stress $\Delta\sigma_{Hr}$ or strain $\Delta\varepsilon_{Hr}$ perturbation is negligible.

To get the regional stress perturbation $\Delta\sigma_{hr}$, we calculate the average of the differences between the reference
and initial stresses. The individual stress differences are given by Eq. (1) in case of stress measurements or Eq. (2) for the
critical stress model. We use the average since the regional stress perturbation is assumed constant across all layers (see
Sect. 2 and Figure 1). For the regional strain perturbation $\Delta\varepsilon_{hr}$, we convert all the stress differences into individual strain
differences, and subsequently compute their average, Eq. (B.4) and Eq. (B.5). The stress-to-strain conversion thus
assumes that the regional strain perturbation $\Delta\varepsilon_{hr}$ equals the average strain contribution of non-coupled layers.

If computation of the regional stress $\Delta\sigma_{hr}$ or strain $\Delta\varepsilon_{hr}$ perturbation is based on stress measurements then likely
only a few points will determine their magnitudes. This is not the case if the critical stress model acts as a reference since
then local differences are available for all layers. This points to a likely tradeoff in the final predictions between paucity
of information (lacking and uncertain measurements) versus potential of systematic errors due to an incorrectly assumed
reference state of critical stress.

### 3.4. Step 4: Updating the initial stress to obtain the final stress profiles

The final step consists in modifying the initial stress profile obtained in step 1, in each layer as a function of the rock
properties, depending on the regional tectonic stress or strain perturbation obtained in step 3, in order to get the final stress
profile that takes tectonic effects into account. We focus on the minimum principal stresses and the calculation of the
intermediate horizontal principal stress is disregarded because the maximum and minimum principal stresses are more
critical to assess fracturing than the intermediate stress. Likewise, according to our assumption, i.e., normal stress regime
and no tectonic perturbation in the direction of the intermediate principal stress, the magnitude of the layer to layer stress
variation is potentially greater in the direction of the minimal principal stress.

We explore both fundamental tectonic models, namely the coupled stress-driven and the strain-driven models,
as applied to the four base predictions developed so far. We obtain therefore a total of 8 different predictive strategies,
leading potentially to 8 different predictive stress profiles (Figure 3). We use the regional stress $\Delta\sigma_{hr}$ and strain $\Delta\varepsilon_{hr}$
perturbations to obtain the final stress profiles assuming respectively a stress- or strain-driven model (Table 1). For
reference, the predictive strategy using an initial uniaxial model, in situ stress measurements and the strain-driven model
is the model most commonly used in the literature [Savage 1992; Blanton and Olson 1999; Beaudoin et al., 2011; Song
et al., 2012].

For the strain-driven model, the local stress corrections $\Delta\sigma_{hl}$ are calculated analytically using the vertical
variation of Poisson's ratio $\upsilon$ and Young's $E$ with Eq. (8). Then, the depth-dependent local stress $\sigma_{hl}$ is calculated with
Eq. (1), the pore pressure $P_p$, and the appropriate initial stresses $\sigma_{hli}$.

For the stress-driven model, the local stress corrections $\Delta\sigma_{hl}$ and depth-dependent local effective stress $\sigma'_{hl}$ are
modelled using a discrete-element method [Cundall, 1988] assuming perfectly coupled layers. We use the same
methodology as described in Roche et al. [2013] and Roche and Van der Baan, [2015]. In summary, we build a perfectly
coupled elastic layered section that mimics the properties of the natural case. An in situ initial state of stress corresponding
to the chosen predictive base strategy (i.e., lithostatic stress $\sigma_l$ or uniaxial stress $\sigma_u$) is set within each element of the
model. A boundary condition equal to the in situ stress plus constant regional tectonic stress perturbation $\Delta\sigma_{hr}$ is applied
to the walls of the model in the direction of the minimum principal stress. Stress transfer across layers is properly
accounted for once the model has reached equilibrium.



Finally the critical stress model can be used to obtain upper and lower bounds for all predicted stresses [Brace and Kohlstedt, 1980]. In the case study below we assume a normal stress regime such that only a lower stress bound is

required since the maximum stresses are determined by the overburden weight. We will also assume a uniform coefficient of friction of 0.6.

## 4. Field case

### 4.1. Study area

The field example is located northeast of Fort Nelson, British Columbia, in the Western Sedimentary Basin of Canada

[Mossop and Shesten, 1994]. The hydrocarbon reservoir (gas) is stimulated by hydraulic fracturing treatments. The studied section includes the Evie, Muskwa and Otter Park low porosity shale members of the Horn River Formation (Figure 4A). They lie above the carbonatic Keg River Formation and under the shale of the Fort Simpson Formation [Curtis et al., 2010; Chalmers et al., 2012]. The shales are overpressurized with a pore pressure gradient of 11 to 16 MPa/Km [Hurd and Zoback 2012].

The rock densities and the compressional and shear wave velocities are obtained from well data (Figure 4B). The depth dependence of the dynamic elastic parameters, Young's modulus $E_d$ and Poisson's ratio $\upsilon_d$, are calculated from the density $\rho$ and the compressional and shear wave velocities, $V_p$ and $V_s$ (Figure 4D,E). The methodology is provided in Roche and Van der Baan [2015].

For practical purposes, in rocks mainly composed of shale, like in this study, there is no difference between the

dynamic and static Poisson's ratio [Mullen et al., 2007]. The static Young's modulus $E_s$ is obtained from the dynamic moduli $E_D$ measured in the well logs using [Mullen et al., 2007]:

$$E_s = E_D\left(0.8 - \phi\right), \tag{11}$$

where $\phi$ is the total porosity of the rocks. For the porosity of the rocks, we use an average value equal to 2%, although it slightly varies between formations. The shale rocks of the Horn River and Fort Simpson Formations are anisotropic with

transverse isotropic properties [Khan et al., 2011]. This anisotropic effect is disregarded here in order to simplify calculations. Also, visco-elasticity likely occurs in some layers, but this behaviour is not considered. In practice, visco-elasticity implies that the long term asymptotic solution can be evaluated through a linear model [see discussion in Roche et al., 2013]. In this case the elastic constants evaluated in this study are likely overestimate, especially in the shale layers.

### 4.2. Stress regime

The Western Canadian Basin is stressed tectonically because of the Rocky Mountains to the southwest, which results in a net difference in the horizontal stresses [*Beaudoin et al., 2011; Reiter et al., 2014*]. The orientation of the maximum horizontal stress is well defined all over the basin and trends NW-SW [Bell and Gough 1979; Bell and Bachu, 2003; Reiter et al., 2014], which is normal to the Rocky mountain trench and folding. However, the tectonic regime is not well delineated by stress measurements [Reiter et al., 2014]. Studies highlight spatial variation in stress regime, with thrust

faulting in the foothills, strike slip within the basin, and normal faulting regime in the eastern part of the basin [Bell and Gough 1979; Bell and Bachu, 2003; Reiter and Heidbach, 2014].

Likewise, depth variations in stress regime are described, with thrust faulting at shallow depth (i.e., <350–600m), strike slip at intermediate depth 500–2500 m, and normal faulting at greater depths >2500m [Fordjor et al., 1983;





McLellan 1987; Reiter and Heidbach, 2014]. In most of the measurements performed in the Basin, the minimum principal
stresses are lower than the vertical stresses calculated from the weight of the overburden rock [Bell and Grasby 2012].
This precludes a thrust fault regime. In the whole Basin, the minimum principal stress gradient ranges from 12 to 27
MPa/Km$^{-1}$, with an average of 18 MPa/Km$^{-1}$ [Bell and Grasby 2012]. The SHmax/Shmin ratio is about 1.3–1.6 [Fordjor
et al., 1983].

More locally in the studied area, the bottomhole instantaneous shut-in pressures do not show consistent variation
from the toe to the heel with minimum principal stresses equal to 26 ±7 MPa and 38 ±8 MPa in the Muskwa and the Evie
members, respectively (Figure 4). This corresponds to a minimum principal stress gradient equal to 13 MPa.Km$^{-1}$ on
average, which is low compared to values measured in a nearby site (i.e., ≈21 MPa/Km$^{-1}$) [Hurd and Zoback 2012], or in
the Western Canadian Basin [Bell and Grasby 2012]. Yet it is consistent with the stress field map provided by Bell and
Grasby [2012].

The minimum principal stress is thus lower than the vertical stress. In addition, the ratio of the minimum principal
stress and the vertical stress is very close to the ratio for a critical state of stress. Thus, it is very unlikely that the maximum
horizontal stress is higher than the vertical stress because it would involve an overcritical state of stress. Our data therefore
indicate that a normal or transtensional stress regime is more likely than a strike slip regime. This may be due to regional
strike-slip faults that induce significant stress perturbations responsible for local normal faulting.

## 5. Results

### 5.1. Initial stresses and reference stresses (steps 1 and 2)

The uniaxial stress $\sigma_u$ and the lithostatic stress $\sigma_l$ are computed using the methodology described in section 3.1. Both
stresses increase with depth due to the weight of the overburden rock (Figure 5A). The increase is slightly lower for the
uniaxial stress $\sigma_u$ than for the lithostatic stress $\sigma_l$. The lithostatic stress $\sigma_l$ increases linearly with depth, whereas layer to
layer variations in uniaxial stress $\sigma_u$ occur due to the vertical variation in the Poisson's ratio $\upsilon$.

For the reference stresses (see Sect. 3.2), the minimum horizontal critical stress $\sigma_{3c}$ increases linearly with depth,
due to the weight of the overburden rock (Figure 5A). The increase is slightly lower than for the lithostatic stress. The
three values of minimum principal stress, as measured in the Muskwa and the Evie members, are indicated in Figure 5.
They are set to be equal to the average of the instantaneous shut in pressures for the different stages along the well. The
resulting value is generally considered to be a proxy for the minimum principal stress [*Zoback, 2010*].

### 5.2. Stress and strain regional perturbations

By comparing the initial and reference values of stress, we obtain the different regional tectonic perturbations $\Delta\varepsilon_{hr}$ and
$\Delta\sigma_{hr}$ (Sect. 3.3). The uniaxial stress $\sigma_u$ is higher than the in situ measurements (Figure 5A). This corresponds to negative
local stress adjustments that are equal to -14 MPa and -2 MPa, in the Muskwa and the Evie members, respectively. This
involves extensional stress and strain regional tectonic perturbations $\Delta\sigma_{hr}$ and $\Delta\varepsilon_{hr}$ equal to -6 MPa and -0.17 mm/m (the
minus sign indicates extension). These are the basis for predictive strategies 5 and 1 in Table 1.

The lithostatic stress $\sigma_l$ is also higher than the in situ measurements (Figure 5A). Since the lithostatic stress $\sigma_l$ is
even higher than the uniaxial stress $\sigma_u$, the resulting stress and strain regional perturbations $\Delta\sigma_{hr}$ and $\Delta\varepsilon_{hr}$ are higher for
the predictive strategies based on the lithostatic stress and equal -34 MPa and -0.94 mm/m. The lithostatic stress $\sigma_l$ is



higher than the minimum horizontal critical stress $\sigma_{3c}$. The resulting stress and strain regional perturbations $\Delta\sigma_{hr}$ and $\Delta\varepsilon_{hr}$ are equal to -21 MPa and -0.75 mm/m, respectively. These are the basis for predictive strategies 6 and 2 in Table 1.

     The uniaxial stress $\sigma_u$ is very close or slightly higher than the critical stress $\sigma_{3c}$ in some formations. The resulting stress and strain regional perturbations $\Delta\sigma_{hr}$ and $\Delta\varepsilon_{hr}$ are thus very small, i.e., 1 MPa and 0.006 mm/m, and compressive (predictive strategies 7 and 3 in Table 1).

For nearly all the predictive strategies the regional tectonic perturbations $\Delta\sigma_{hr}$ and $\Delta\varepsilon_{hr}$ are extensional. The exception is the predictive strategy that is based on the uniaxial and the critical models, where the tectonic perturbation is compressive. However, this perturbation is so low that its effect is negligible. This case is not illustrated in figure 5. These results show that a normal stress regime does not involve necessarily extensional tectonic perturbations. A compressive perturbation can also result in a normal stress regime if it is applied on a low magnitude initial state of stress,

like in the case of a uniaxial state of stress. The effect of the various regional tectonic perturbations on the final predicted stresses is presented in the next sub-section.

**5.3. Local stress corrections and predicted stress profiles**

     Next we compute the local stress corrections $\Delta\sigma_{hl}$ and the final stress profiles (see Sect. 3.4). In this section we present first the results obtained for the strain driven model, then for the stress driven model. In the next sub-section we address

the observed differences between both models.

     The local stress corrections $\Delta\sigma_{hl}$ are negative for an extensional regional tectonic strain or stress perturbation $\Delta\varepsilon_{hr}$ and $\Delta\sigma_{hr}$ (Figure 5). For a strain-driven model, the local stress corrections $\Delta\sigma_{hl}$ are positively correlated to the Young's modulus $E$, their absolute values are therefore greater in the Keg River Formation, followed by the Horn River and finally the Fort Simpson Formations (Figure 5B, C and D). The magnitude of the local stress corrections $\Delta\sigma_{hl}$ in a specific layer

also depends on the magnitude of the regional strain perturbations $\Delta\varepsilon_{hr}$. They are maximal for strategy 2 based on the lithostatic model and the in situ stress measurements.

     The predicted minimum principal stress $\sigma_{hl}$ depends on the initial state of stress and the local stress corrections $\Delta\sigma_{hl}$. Because the local stress corrections $\Delta\sigma_{hl}$ are negative, the predicted minimum stress $\sigma_{hl}$ decreases due to inclusion of the tectonic effects. For all prediction strategies, due to the variation in Young's modulus $E$, the decrease is most

important in the Keg River Formation, followed by the Horn River and finally Fort Simpson Formations (Figure 5B, C and D). Likewise, the decrease is slightly lower in the Otter Park member than in the surrounding Muskwa and Evie members. Consequently, in regard to the Muskwa and Evie target members: there is a negative change in the minimum horizontal stress at the lower interface with the Keg River Formation; and a positive change at the upper interface with the Fort Simpson Formation; the intermediate Otter Park member exhibits a higher stress than the surrounding targets

Formations (Figure 5 B, C and D).

     The predicted minimum stress $\sigma_{hl}$ also depends on the magnitude of the regional strain perturbations $\Delta\varepsilon_{hr}$. The changes at formation interfaces are maximal for strategy 2 based on the lithostatic model and the in situ stress measurements. In this case, the layer to layer stress variation reaches up to 13 MPa between the Horn River and the Keg River Formations, 10 MPa between the Horn River and the Fort Simpson Formations and 3 MPa between the Otter Park

member and the surrounding Muskwa or Evie members (Figure 5D).

     For the predictive strategies 1 and 3, based on the uniaxial model (Table 1), layer to layer stress variations are initially present before the tectonic perturbation due to the vertical variation in the Poisson's ratio (Figure 5B). Depending




on the rock properties, the local stress corrections $\Delta\sigma_{hl}$ may promote the initial layer to layer stress variations, e.g. between the Horn River Formation and the Fort Simpson Formation. In other instances the local stress corrections $\Delta\sigma_{hl}$ may inhibit

them, e.g., between the Horn River Formation and the Keg River Formation. Depending on the magnitude of the local stress corrections $\Delta\sigma_{hl}$, the layer to layer stress variations in this initial stress profile may be decreased or enhanced by the computed local stress corrections. Nevertheless, in most of the models, the local stress corrections $\Delta\sigma_{hl}$ are higher than the initial variations. This implies that the tectonic effects can outweigh or reverse any initial stress variations due to the Poisson's effect.

For the stress-driven model, similar local stress corrections $\Delta\sigma_{hl}$ and minimum horizontal stress $\sigma_{hl}$ trends and magnitudes are obtained as for the strain-driven model (compare Figures 5B-D with 5F-H, as well as Figure 6). Similarly to the strain-driven model, from the Muskwa and Evie target members: the minimum horizontal stress decreases at the interface with the Keg River Formation and increases at the interface with the Fort Simpson Formation and the Otter Park member exhibits a higher stress (Figures 5F-H).

The predicted magnitudes of the layer to layer stress variations are also very similar. The maximum values are obtained for prediction strategy 8, where layer stress variations reach respectively 11 MPa between the Horn River and the Keg River Formations, 8 MPa between the Horn River and the Fort Simpson Formations and 5 MPa between the Otter Park member and the surrounding Muskwa or Evie members (light grey curves in figure 5C). In this case, however, predicted stress changes are due to stress transfer between layers and depend on the contrast in Young's moduli between

layers and their thicknesses (see Appendix B). This causes important differences between the stresses predicted with the stress or strain-driven models. This is explored in the next sub-section.

**5.4. Comparison of stresses predicted with the strain and stress–driven models**

Almost no regional tectonic perturbations $\Delta\sigma_{hr}$ and $\Delta\varepsilon_{hr}$ occur for prediction strategies 3 and 7 based on the initial uniaxial model and the critical model. In this case, the final predicted stress profiles are visually similar for both the stress- and

strain-driven models because they remain essentially equal to the uniaxial stress profile.

     For the other strategies, the results obtained with the stress- and strain-driven models show similar trends (Figure 6). For both models, the magnitude of the local stress corrections depends on the magnitude and polarity (i.e., compression or extension) of the tectonic perturbations $\Delta\varepsilon_{hr}$ and $\Delta\sigma_{hr}$. Depending on the tectonic perturbation, the resulting local stress corrections $\Delta\sigma_{hl}$ may promote, maintain or inverse the initial layer to layer variations in the stress uniaxial profile due to

the depth dependence of the Poisson's ratio. Yet, we note two significant differences between the models.

     First, for the strain-driven model, the local stress corrections $\Delta\sigma_{hl}$ are constant within a layer, whereas they are concentrated near the formation interfaces for the stress-driven model. As a consequence, the final local stress gradient for the strain-driven model remains equal to the gradient of the initial state of stress. In the stress-driven model the final predicted stress gradient changes within a layer, as well as between layers. The gradient may be higher or lower than the

stress gradient of the initial state of stress.

     Second, for the stress-driven model, the local stress corrections $\Delta\sigma_{hl}$ fluctuate around the regional perturbation $\Delta\sigma_{hr}$, i.e., the local stress correction averaged over several layers equals the regional perturbation. This average value is independent of the layering. For instance, for strategy 6 displaying the largest regional perturbation, the average local stress corrections calculated over the Fort Simpson and the Horn River Formations equals -33 MPa and it is -34 MPa over

the Horn River Formations and the Keg River Formations.



Conversely, for the strain model, the average magnitude varies as a function of the elastic properties of the studied section. For instance, very different average local stress corrections are found for strategy 2, i.e., -23 MPa and -40 MPa for the Fort Simpson-Horn River Formations and the Horn River-Keg River Formations, respectively.

As a consequence, a strain-driven model may exhibit lower stress corrections than a stress-driven model in the compliant layers and higher local stress corrections occur for the strain-driven model in the stiffer lithologies. In the case of an extensional perturbation, like in the study case, the predicted minimum horizontal stress $\sigma_{hl}$ is thus higher in compliant layers, e.g., the Fort Simpson Formation, and it is lower in stiff ones, e.g. the Keg River Formation, for the strain models. The horizontal stresses become very similar in the Horn River Formation that has a mean stiffness and where the tectonic perturbation has been originally calibrated (Figure 5).

**5.5 Comparison of the stress predictions with a critical state of stress**

On average, the predicted minimum principal stress $\sigma_h$ is close to the critical state of stress for all predictive strategies, expect for strategy 6 (Figure 5), implying generally consistent results. It is worth noting that the regulator has declared that hydraulic fracturing in this area has led to human-induced seismicity *[BCOGC, 2012, 2014]*. Most of the larger magnitude events occurred within the Keg River formation or below. The prediction of a critically stressed area is thus

also reasonable. A few layers are predicted to be slightly infra-critically stressed (i.e. $\sigma_{3c} > \sigma_h$) but more commonly, layers are supra-critically stressed (i.e. $\sigma_{3c} < \sigma_h$). This may indicate that the minimum principal stress $\sigma_h$ is locally underestimated.

A closer scrutiny of the various stress profiles in Figure 5 shows that the strain-driven tectonic models (strategies 1-4) predict that the largest supra-critical stresses occur in the Keg River formation, whereas the stress-driven models

(strategies 5-8) show also supra-critical stresses above this formation. A critically stressed Keg River formation is more likely given that the induced seismicity occurred within or below this formation.

Finally predictions 2 and 6 based on an initial lithostatic stress and the in-situ stress measurements show the most supra-critical stresses independent of the chosen tectonic model. This can be explained by the fact that the in-situ stress measurements are substantially lower than the critical stresses. Furthermore, using the lithostatic stress as the initial state

produces higher tectonic effects than using any other initial stress state. As a consequence, stress predictions based on the lithostatic stress and in-situ stress measurements creates strongly supra-critical predictions in this case.

**5.6. Uncertainty in stress prediction**

The various inputs/models give rise to inherent uncertainty in stress prediction. In the study case the maximum differences in final stress obtained between all the stress predictions reach 20 MPa, 18 MPa and 28 MPa in the Fort Simpson, Horn

River, and Keg River Formations, respectively. For the different stress predictions that are based on a uniaxial stress, the maximum difference reaches 7 MPa, 6 MPa and 8 MPa in the Fort Simpson, Horn River, and Keg River Formations. For an initial lithostatic stress, they reach 20 MPa, 15 MPa and 28 MPA in the same Formations. When using the critical stress as reference stress, the maximum changes between the various stress predictions are 6 MPa, 9 MPa and 16 MPa in the Fort Simpson, Horn River, and Keg River Formations. They are 15 MPa, 4 MPa and 16 MPa in the same Formations

when using the in situ stress measurement as reference stresses. Assuming a strain-driven model, they reach 8 MPa, 8 MPa and 18 MPa and 14 MPa, 18 MPa and 12 MPa assuming a stress driven model. Likewise, the range of predicted stress discontinuities between layers are 1-7 MPa and 1-12.5 MPa between the Fort Simpson and the Horn River



Formations and between the Horn River and the Keg River Formations, respectively. These uncertainties in stress prediction have fundamental implications on fracturing and containment capacity.

**Discussion**

In situ stress depends on several effects, including pore pressure, rock viscosity, rock anisotropy, preexisting fracturing, thermal effects, etc. [Cornet et Burlet 1992; Addis et al., 1996; Khan et al., 2011; Meixner et al., 2014; Roche and Van der Baan, 2015]. In this paper we focus on tectonic forces in combination with the vertical variation in rock properties while disregarding other potential effects. First, we discuss the choice of the initial and the reference stress, then the stress-

and the strain-driven model, and finally uncertainties in stress prediction. The choice of the initial regional tectonic regime and whether tectonic perturbations in one direction or in both horizontal directions are taken into account are also critical, but these are beyond the scope of the paper.

**6.1. Initial state of stress**

Stress predictions are based on an initial stress model that is modified to account for the tectonic effects. We used two

models for the initial stress: the lithostatic and uniaxial models. In the literature most stress predictions are based on an initial uniaxial model [Voight and St. Pierre, 1974; Haxby and Turcotte, 1976; Savage 1992; Blanton and Olson 1999], and few are based on an initial lithostatic model [McGarr 1988; Roche et al, 2014; Roche and Van der Baan, 2015]. The choice of the initial stress model influences on the final stress predictions for several reasons.

For a uniaxial model, initial layer to layer stress variations occur due to Poisson's effect. These variations are

added to those created by tectonic effects. In the study case, the tectonic effects dominate Poisson's effects. Still, Poisson's effect may change the magnitude and direction of the final layer to layer stress variations if the tectonic effects are low, the variation in Poisson's ratio is significant and the pore pressure is low.

The magnitude of the regional tectonic perturbation depends on the chosen initial stress model. For instance, the initial stresses calculated with a uniaxial model are lower than those calculated with a lithostatic model. Hence, for an in

situ stress measurement that is higher than the lithostatic stresses (i.e. compressive regime), the magnitude of the tectonic perturbations is greater for an initial uniaxial model than for a lithostatic model. As a consequence, stress predictions based on a uniaxial model tend to underestimate the layer to layer stress variations. For an extensional regime, the tectonic perturbation may be either greater or smaller for the lithostatic model, if the in situ stress measurement is closer to the lithostatic state of stress, or to the uniaxial stress, respectively.

The choice of the initial model is potentially more critical when looking at its effect on the polarity of the regional tectonic perturbations. For instance, perturbations that are in compression or in extension may be obtained if the in situ stress measurements are comprised between the lithostatic and uniaxial stresses. In such a case, by assuming a tectonic perturbation in one direction, final stress predictions based on an initial uniaxial stress are likely to overestimate and underestimate the minimum stress in the stiff and compliant layers, respectively. Consequently, tensile and shear failures

are more likely to appear inhibited in the stiff layer and promoted in the compliant layer.

This discussion raises the question: Which model is more likely to dominate? The stress state predicted by the uniaxial model has rarely been observed in field data [Jaeger et al., 2009 and McGarr 1987, 1988]. Also, it creates a bias toward highly extensional tectonic regimes, and excludes thrust and strike-slip regimes [McGarr 1987, 1988]. However, the lithostatic model also appears implausible. There is little experimental evidence to support this model because the

time dependent failure mechanisms that could remove all the deviatoric stresses for indefinite periods of time do not seem



to exist in the upper crust [Kirby and McCormick 1984; Mc Garr 1988]. Nevertheless, the lithostatic stress state appears the more realistic reference stress [Mc Garr 1988].

**6.2 Reference state of stress**

The initial stresses influence the computed tectonic perturbations. Their magnitudes and polarities also depend on the reference stresses. Two types of reference are used here: in situ stress measurements and critical stresses. In-situ stress measurements are a better choice because they provide a direct measurement of the local stress. However, such data are often not available, have uncertainty, or are limited to specific depths. Thus, it is useful to be able to predict stress based only on models. This can be done using the critical state of stress. In the latter case, we assume that the stress is controlled by friction on preexisting faults. However, it has been shown that the stress observed close to an active faults implies a

plastic behaviour for the fault gouge that does not satisfy Coulomb failure criterion, but rather obeys a non-associative plastic flow rule [Sulem, 2007]. In this case, we may expect a lower critical stress than the one used in this paper.

In this study, the difference between the reference stresses, i.e. in-situ measurement and critical stress, is relatively small, compared to the difference between the initial stresses, i.e. lithostatic and uniaxial stresses. Hence, the change in tectonic perturbation, obtained for a similar initial stress but different reference stresses, is only 25% of the

change obtained for similar reference stress but different initial stresses. In our case, the stress predictions obtained using in-situ measurement and critical stresses are thus similar.

**6.3. Strain-driven models versus stress-driven models**

For similar initial and reference stresses, the stress predictions obtained with the strain-driven and the stress-driven models share a similar trend, consistent with in situ stress measurements, as a first approximation. For both models, stresses

predicted in one specific layer depend on the tectonic perturbation and the elasticity. Nevertheless, significant differences occur in terms of stress magnitude and stress gradients (see Sect. 5.4). It is therefore fundamental to know which model dominates in nature, or if both models occur.

The strain-driven model has been widely used, notably as a standard method for oil and gas reservoir exploration [Thiercelin and Plumb, 1994; Blanton and Olson, 1999; Beaudoin et al., 2011; Song and Hareland, 2012]. The stress-

driven model may involve discontinuous strain across layer boundaries if the layers are not coupled together. Such a behavior may appear as a nonphysical consequence of the model leading authors to disregard this model in favor of the strain-driven model [Blanton and Olson 1999]. This explains why the stress-driven model is scarcely used [Teufel and Clark, 1984; Bourne, 2003; Roche et al., 2013]. However, if the layers are coupled together, both the regional stress and regional strain are continuous throughout layering. Likewise, the possible occurrence of strain discontinuities does not

appear decisive in disregarding the stress-driven model, because strain decoupling likely exists in natural rocks [Cornet et Burlet 1992; Meixner et al., 2014]. Also, a recent study shows that the fracturing depends on the contrast of elasticity between layers, as well as the layer thicknesses, rather than solely on the properties of the layer, in which fracturing develops [Roche et al., 2014]. This tends to support the stress-driven model, rather than the strain-driven model. Lastly, bed-parallel faults occurring in tabular rocks have been described [Roche et al., 2012a and b]. Such a structure may

highlight strain discontinuities. This is also in accordance with the stress-driven model. Otherwise, for the strain-driven model, an additional mechanism is involved to create these structures.



### 6.4. Final remarks: Uncertainty in stress predictions

Accurate prediction of the in situ stresses in the Earth is hampered by epistemic and aleatoric uncertainty. Epistemic (or systematic) uncertainty is caused by lack of knowledge and data; aleatoric (or statistical) uncertainty is caused by the inherent randomness of a phenomenon such as the rolling of dice. The variation in final predicted stresses stems from a range of different sources for uncertainty, including observational and interpolation/extrapolation uncertainty, parameter uncertainty, model uncertainty and numerical/algorithmic uncertainty [*Kennedy* and O'Hagan, 2001].

Observational, interpolation and extrapolation uncertainty arises for instance since the stress measurements used in step 2 are prone to observational error, thus influencing the final stress predictions. In addition, the stress measurements are often limited to specific layers and sites, thereby requiring interpolation and/or extrapolation to derive values appropriate for the areas and depth zones under consideration. This introduces uncertainty and spread in the final predicted stress values.

Next, parameter uncertainties are also important, for instance in assumed elastic parameters (Young's modulus, Poisson's ratio) and friction coefficients. Some of these parameters are derived from well logs and thus prone to observational error; yet even in the absence of such observational uncertainty, we lack knowledge on how to convert measured parameters (e.g., dynamic moduli) exactly to required modeling parameters (e.g., static moduli), the control of visco-elasticity, or the behaviour for the faults, thus creating parameter uncertainty. Furthermore, each predictive strategy depends on different parameters, with their own uncertainties and possibly biases, thus introducing further diversity in the final predicted stresses.

Moreover we have model uncertainty in that full knowledge on the actual driving forces and most appropriate geologic boundary conditions is lacking. For instance, both the stress-driven and strain-driven models are geologically plausible. Likewise, lithostatic, uniaxial and critical states of stress are reasonable assumptions in various circumstances, yet the most appropriate one is rarely known. Also, assumptions implicit in the derivation of the provided equations create model uncertainty. Again, lack of knowledge (epistemic uncertainty) causes diversity in final predictions.

Finally, there is numerical (or algorithmic) uncertainty, caused by numerical errors and approximations when solving for the stress state, e.g., using the discrete-element method in step 4 or in the implementation of any of the provided equations. We assume that these only play a very minor role in our case; for instance, convergence of solutions was closely monitored. Nonetheless, numerical uncertainty remains a possible source of uncertainty in all computer simulations.

### Conclusion

Different strategies to predict the vertical variations in the in situ stresses lead to different answers. Such stress predictions take into account the weight of the overburden rock, the pore pressure, the variation of the rock properties and the tectonic effects. In addition, they assume either stress- or strain-driven models, an initial uniaxial or lithostatic model and use a critical model or in situ stress measurements. The different prediction strategies generally lead to similar trends in predicted stresses; yet differences appear both within a layer and between layers, due to fundamentally different underlying mechanisms, assumptions and governing parameters. The spread and diversity in final stress predictions is caused by mostly epistemic uncertainty, expressed as lack of knowledge, data and/or observational errors in some measurements or variables. Nonetheless, the combined analysis of all eight stress predictions helps reveal the true uncertainty, or conversely similarity, in all stress predictions.



**Acknowledgements**

The authors would like to thank the sponsors of the Microseismic Industry Consortium and the Helmholtz-Alberta
Initiative for financial support and Itasca for software licensing. We also thank A. Gudmundsson, A. McGarr, P.
McLellan, A. Zang and O. Heidbach for helpful comments.

**Appendix A: Lithostatic, uniaxial and critical state of stresses**

In the lithostatic model, the principal stresses $\sigma_1$, $\sigma_2$ and $\sigma_3$ are equal to the lithostatic stress $\sigma_l$ calculated from the
overburden stress as determined by the density $\rho$. Thus,

$$\sigma_1 = \sigma_2 = \sigma_3 = \sigma_l = \sigma_v = \rho g z ,$$  (A.1)

where $\sigma_v$ is the overburden vertical stress, $g$ is the acceleration due to gravity, and $z$ is the depth. This model assumes that
rocks cannot support differential stresses, such that horizontal stresses equalize to the overburden (vertical) stress over a

long period of time, due to inelastic deformations. This model may be used for low viscosity, plastic rocks, or for a
relatively long period of time without modification in external boundary conditions. For instance, a stress field close to
lithostatic has been described in shales [*Warpinski et al., 1985*].

The uniaxial stress model assumes that no regional horizontal strains exist but horizontal stresses are imposed

by the Poisson's effect (that is, a horizontal force due to vertical loading) [*Savage et al., 1992; Jaeger et al., 2009*]. The
overburden stress $\sigma_v$ creates a uniaxial horizontal stress $\sigma_u$ due to the lateral extension of the medium that is impeded by
non-deformable walls (i.e., no lateral strains). In an elastic isotropic rock, we obtain [*Engelder and Fischer 1994; Addis
et al., 1996; Blanton and Olson 1999*]:

$$\frac{\sigma'_u}{\sigma'_v} = \frac{\left(\sigma_u - \alpha P_p\right)}{\left(\sigma_v - \alpha P_p\right)} = \left(\frac{\upsilon}{1-\upsilon}\right),$$  (A.2)

where $P_p$ is the in situ pore pressure, $\upsilon$ the Poisson's ratio, $\alpha$ is the Biot pore-pressure coefficient, and the apostrophe '
indicates an effective stress. With a Poisson's ratio of 0.3, the ratio between the uniaxial and vertical stresses equals
0.43.This model explicitly excludes compressive regimes [*Jaeger et al., 2009; McGarr 1988*]. At best it shows qualitative
agreement with stress measurements, but rarely an accurate prediction while in other cases, stress measurements are in
contradiction to the model [*Warpinski et al., 1985; McLellan, 1987; Whitehead et al., 1987; Ahmed et al., 1991; Plumb

et al., 1991; Thiercelin and Plumb, 1994; Addis et al., 1996*]. Yet, its main advantage is that horizontal stresses are easy
to compute since only knowledge of the density, pore pressure and Poisson's ratios are required.

In the critical stress model, the horizontal stresses are assumed equal but the ratio between horizontal and vertical
stress is set such that pre-existing, optimally-oriented faults and fractures are at the point of shear failure. The ratio of
effective maximum to minimum critical principal stresses $\sigma'_{1c}/\sigma'_{3c}$ then only depends on the frictional strength of the pre-

existing faults as their cohesion is set to zero [*Zoback, 2010*], that is,

$$\frac{\left|\sigma'_{1c}\right|}{\left|\sigma'_{3c}\right|} = \frac{\left|\sigma_{1c} - P_p\right|}{\left|\sigma_{3c} - P_p\right|} = \left(\sqrt{\mu^2 + 1} + \mu\right)^2 ,$$  (A.3)





where μ is the coefficient of friction. For $\mu = 0.6$, the ratio between the critical stresses equals 3.1. An equivalent ratio is obtained with a Poisson's ratio equal to 0.25 for the uniaxial stress model. The stresses calculated using the uniaxial model with Poisson's ratios higher than 0.25 are thus over-critical.

**Appendix B: Coupled stress-driven model**

In order to describe analytically such a behavior, we assume only one regional tectonic perturbation $\Delta\sigma_{hr}$ exists applied in the direction of the minimum principal stress. In such a case, the regional strain continuity between the layers can be expressed with the following equation:

$$\Delta\varepsilon_{hrc} = \Delta\varepsilon_{hrs},$$

(B.1)

where $\Delta\varepsilon_{hrc}$ and $\Delta\varepsilon_{hrs}$ are the regional strain perturbations in the compliant and stiff layers, respectively (Figure 1F). The subscript $c$ and $s$ refer to the compliant and stiff layer, respectively. The strain in each layer can be split into two components, namely a strain correction for the non-coupled case and a second correction due to coupling, producing

$$\Delta\varepsilon_{hrs} = \Delta\varepsilon_{hsnc} + \Delta\varepsilon_{hst}, \text{ and } \Delta\varepsilon_{hrc} = \Delta\varepsilon_{hcnc} + \Delta\varepsilon_{hct}$$

(B.2)

where $\Delta\varepsilon_{hsnc}$ and $\Delta\varepsilon_{hcnc}$ are the local strain corrections obtained for non-coupled layers, $\Delta\varepsilon_{hst}$ and $\Delta\varepsilon_{hct}$ are the local strain

corrections, induced by the coupling. The latter lead to strain transfer denoted by the subscript $t$. Combining Eq. (B.1) and Eq. (B.2), we obtain

$$\Delta\varepsilon_{hcnc} + \Delta\varepsilon_{hct} = \Delta\varepsilon_{hsnc} + \Delta\varepsilon_{hst},$$

(B.3)

The various strain corrections can be computed from Eq. (10) by assuming a single strain perturbation in one horizontal

direction occurs, with zero stress and zero strain in the second horizontal direction, producing

$$\Delta\varepsilon_{hst} = \frac{\left(1 - \upsilon_s^{\,2}\right)}{E_s}\Delta\sigma_{hst} \text{ and } \Delta\varepsilon_{hsnc} = \frac{\left(1 - \upsilon_s^{\,2}\right)}{E_s}\Delta\sigma_{hr},$$

(B.4)

in the stiff layer, and in the compliant layer:

$$\Delta\varepsilon_{hct} = \frac{\left(1 - \upsilon_c^{\,2}\right)}{E_c}\Delta\sigma_{hct} \text{ and } \Delta\varepsilon_{hcnc} = \frac{\left(1 - \upsilon_c^{\,2}\right)}{E_c}\Delta\sigma_{hr},$$

(B.5)

where $\Delta\sigma_{hct}$ and $\Delta\sigma_{hst}$ are the local stress transfers in the compliant and stiff layers, respectively. Using Eq. (B.3 - B.5),

we obtain

$$\frac{\left(1 - \upsilon_s^{\,2}\right)}{E_s}\Delta\sigma_{hr} + \frac{\left(1 - \upsilon_s^{\,2}\right)}{E_s}\Delta\sigma_{hst} = \frac{\left(1 - \upsilon_c^{\,2}\right)}{E_c}\Delta\sigma_{hr} + \frac{\left(1 - \upsilon_c^{\,2}\right)}{E_c}\Delta\sigma_{hct},$$

(B.6)

The equilibrium condition sets that there is zero net force across a plane that is normal to the layers [*Holzhausen and Johnson 1979; McGarr 1988*]:





$$\Delta\sigma_{hst}h_s + \Delta\sigma_{hct}h_c = 0, \tag{B.7}$$

where $h_s$ and $h_c$ are the stiff and compliant layers thicknesses, respectively. Using Eq. (B.6) and Eq. (B.7), the local stress transfers in the stiff and compliant layers become:

$$\frac{h_s(A-B)\Delta\sigma_{hr}}{h_s B + h_c A} = \Delta\sigma_{hct} \text{ and } \Delta\sigma_{hst} = -\frac{h_c}{h_s}\Delta\sigma_{hct}, \tag{B.8}$$

with

$$A = \frac{\left(1 - \upsilon_s^{\ 2}\right)}{E_s} \text{ and } B = \frac{\left(1 - \upsilon_c^{\ 2}\right)}{E_c}, \tag{B.9}$$

645        The local tectonic stress corrections then correspond to the sums of the local stress transfers due to coupling and the regional stress perturbations. That is,

$$\Delta\sigma_{hl_s} = \sigma_{hr} + \Delta\sigma_{hst} \text{ and } \Delta\sigma_{hlc} = \sigma_{hr} + \Delta\sigma_{hct}, \tag{B.10}$$

Finally, we can calculate the local stress in each layer, with the following equations:

$$\sigma'_{hl_c} = \sigma_{hi} - P_p + \sigma_{hr} + \Delta\sigma_{hct}, \tag{B.11}$$

$$\sigma'_{hl_s} = \sigma_{hi} - P_p + \sigma_{hr} + \Delta\sigma_{hs_t}, \tag{B.12}$$

Eq. (B.11-B.12) hold for periodic media composed of two layers with Young's moduli $E_s$ and $E_c$ and thickness $h_s$ and $h_c$, where all layers are coupled.

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

**Table 1.** The 8 predictive strategies[a]

| predictive strategies | initial models | reference models | tectonic models | $\Delta\varepsilon_{hr}$ (mm/m) [b] | $\Delta\sigma_{hr}$ (MPa) [c] |
|---|---|---|---|---|---|
| 1 | uniaxial | measurements | strain-driven | -0.17 | - |





| | | | | | |
|---|---|---|---|---|---|
| 2 | lithostatic | | | -0.94 | - |
| 3 | uniaxial | critical | | 0.006 | - |
| 4 | lithostatic | | | -0.75 | - |
| 5 | uniaxial | measurements | | - | -6 |
| 6 | lithostatic | | stress-driven | - | -34 |
| 7 | uniaxial | critical | | - | 1 |
| 8 | lithostatic | | | - | -21 |

[a]: *in bold, the predictive strategy commonly used in literature*
[b]: *regional tectonic strain perturbation*
[c]: *regional tectonic stress perturbation*

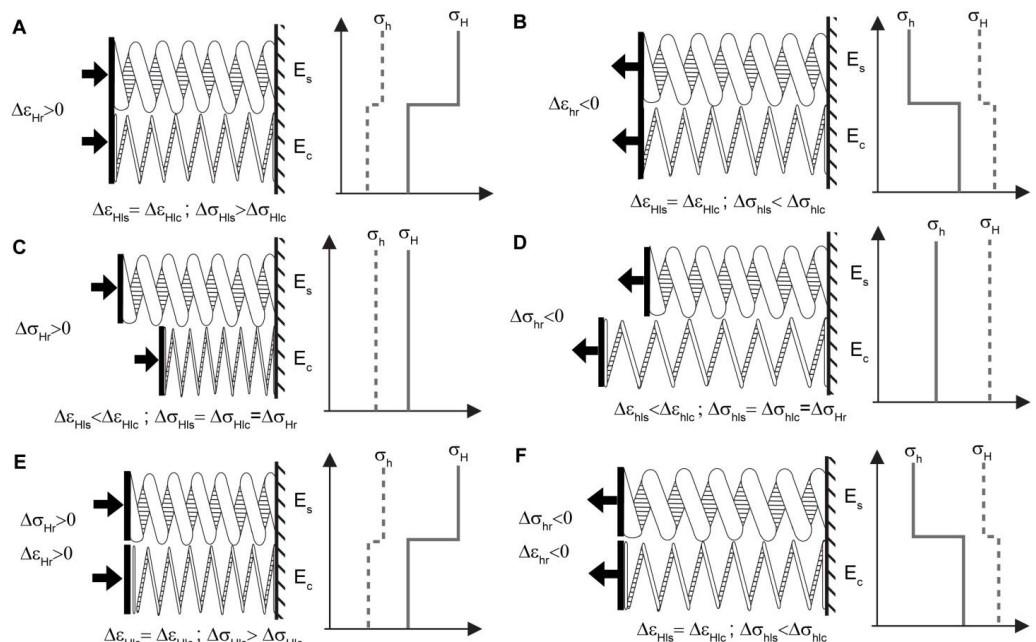

**Figure 1:** Conceptual representation of the stress and strain-driven models. A to F: stiff layer on top with Young's
modulus, $E_s$ and compliant layer underneath with Young's modulus, $E_c$, represented by thick and thin springs,
respectively. Dashed wall is non-deformable (fixed). A single regional perturbation occurs in the direction of the
compression (A, C and E), or extension (B, D and F). A and B: strain-driven model with imposed regional strain
perturbation that is positive (shortening), $\Delta\varepsilon_{Hr,}$ or negative (i.e., lengthening) $\Delta\varepsilon_{hr}$. C-F: imposed regional stress
perturbation that is positive (i.e., compressional), $\Delta\sigma_{Hr,}$ or negative (i.e., extensional) $\Delta\sigma_{hr}$. C and D: non-coupled stress-
driven model; E and F: fully coupled stress-driven model. Each subfigure shows the boundary conditions (next to the
bold horizontal arrows), a conceptual spring diagram, the resulting local stress and strain corrections (underneath the
springs), and the resulting final local stresses in each layer. Subscripts $h$ and $H$: minimum and maximum principal stresses
in the direction of the extension and compression, respectively. Subscripts $c$ and $s$: compliant and stiff layers. $\Delta\sigma_{hls}$,
$\Delta\sigma_{Hls}$, $\Delta\sigma_{hlc}$, $\Delta\sigma_{Hlc}$ : local stress corrections; $\Delta\varepsilon_{hs}$, $\Delta\varepsilon_{Hs}$, $\Delta\varepsilon_{hc}$, $\Delta\varepsilon_{Hc}$ : local strain corrections resulting from the regional




tectonic perturbations, $\Delta\sigma_{hr}$, $\Delta\sigma_{Hr}$, $\Delta\varepsilon_{hr}$, $\Delta\varepsilon_{Hr}$. Local stress equalization happens in the non-coupled stress-driven models
      (C and D). Both strain-driven and coupled stress-driven models can lead to similar local stress profiles, but with very
      different boundary conditions (A, B, E and F).

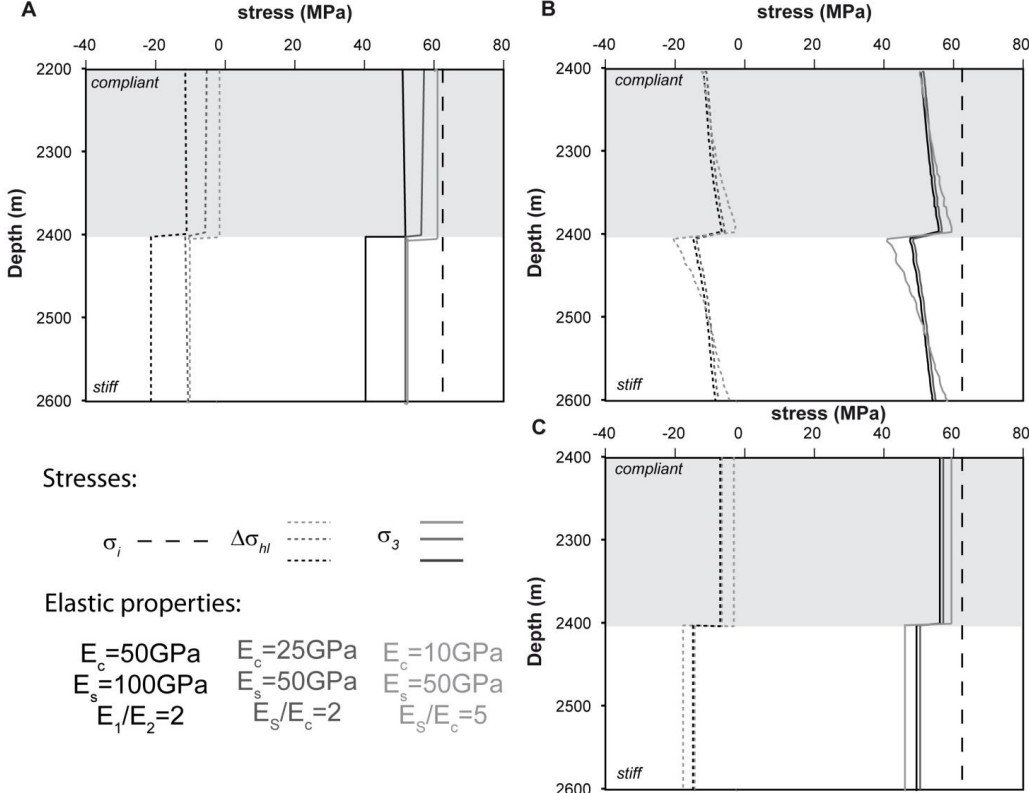

**Figure 2:** Examples of stress predictions, $\sigma_3$, and local stress corrections, $\Delta\sigma_{hl}$, in the strain (A) and stress-driven models
(B and C). A- C: bi-layer media of a 200 m thick compliant layer (Top) with Young's modulus $E_c$ and a 200 m thick stiff
      layer (bottom) with Young's modulus $E_s$. Various combinations of Young's moduli are used as indicated in the figure.
      The Poisson's ratio is constant at 0.25. A: a regional strain perturbation is imposed of 0.002 m/m shortening. Stresses are
      calculated using the analytic solution described in section 2.5. B and C: a regional stress perturbation of -10 MPa is
      imposed. Stresses are calculated using the numerical (B) and analytic solution (C) as described in sections 2.6 and 2.7.
The initial horizontal stress is the lithostatic stress ($\sigma_i=\sigma_l$) calculated using an average 2400 depth. Local stress corrections
      ($\Delta\sigma_{hl}$) and the minimum horizontal stress ($\sigma_3$): dotted and continuous lines and shades of grey for indicated combinations
      of Young's moduli.



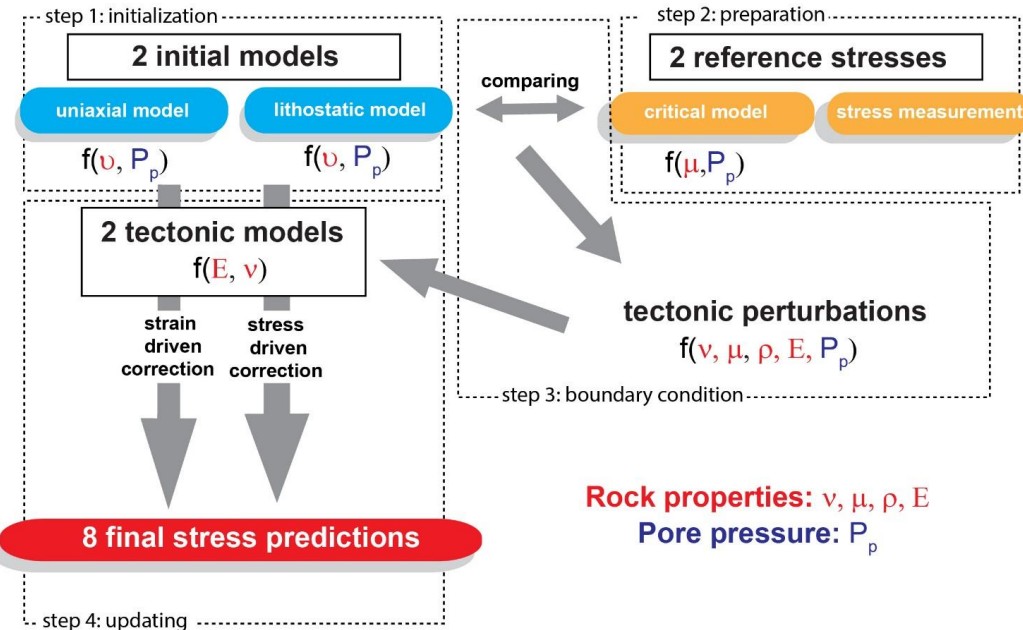

**Figure 3:** Flow chart detailing the different prediction strategies. See section 3 for details. Rock properties used in the different models: $\upsilon$ Poisson's ratio, $\mu$ friction, $\rho$ density, $E$ Young's modulus.

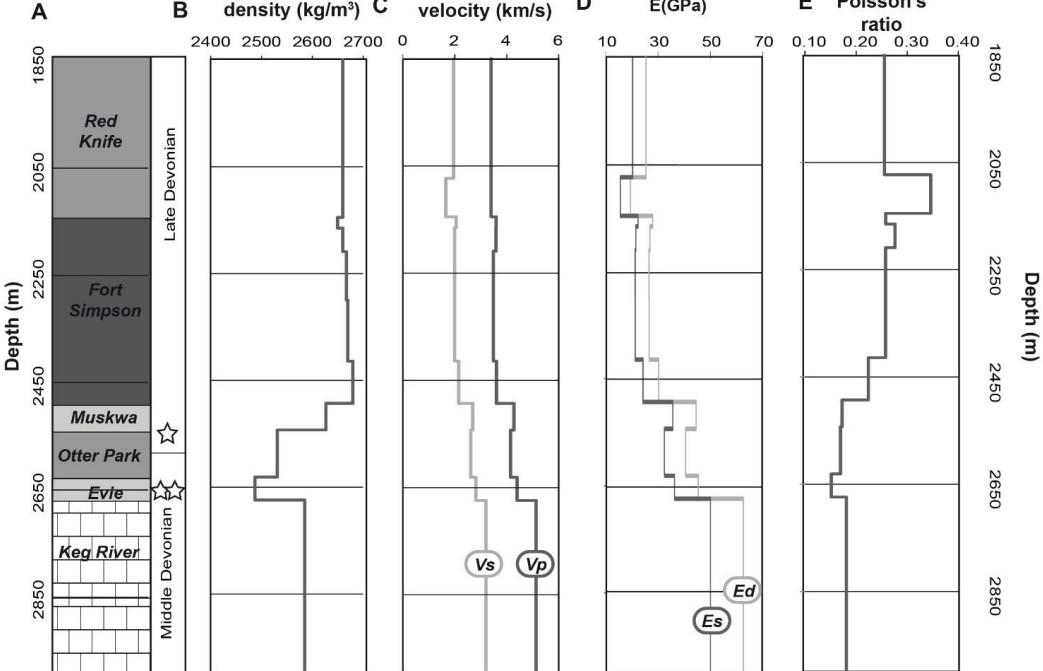

**Figure 4:** Petrophysical and mechanical properties of the study case. A: Lithological section: Fort Simpson, Muskwa, Otter Park and Evie members are clay-rich rocks represented in various shades of grey. The Keg River Formation is





carbonate. The white stars represent the injection levels. B and C: Depth dependence of the density and the P- and S-wave velocities derived from the well data. D and E:variation of respectively the Young's Modulus and Poisson's ratio with depth, calculated from the density and velocities. In D, the dynamic and static Young's modulii are indicated.





**Figure 5:** Calculated stresses for the various models and predictive strategies. A: uniaxial ($\sigma_u$), lithostatic ($\sigma_l$) and critical

($\sigma_{3c}$) models. B-G: Stress predictions for the various predictive strategies including the local stress corrections $\Delta\sigma_{hl}$ and

the predicted minimum principal stress $\sigma_{hl}$. Circled numbers: labels of the predictive strategies as presented in table 1.

The relevant initial stress (i.e., uniaxial $\sigma_i$ or lithostatic $\sigma_u$) and reference stresses (i.e., in situ stress measurements and

the critical stress $\sigma_{3c}$) are also represented for each predictive strategy. E and I: Simplified stratigraphic column of the

studied area with the same legend as for Figure 4. The diamonds represent the average values of the minimum principal

stress (i.e., instantaneous shut in pressure values). All the calculations assume a pore pressure gradient of 13 MPa/km and

an average density of 2600 kg.m$^3$.

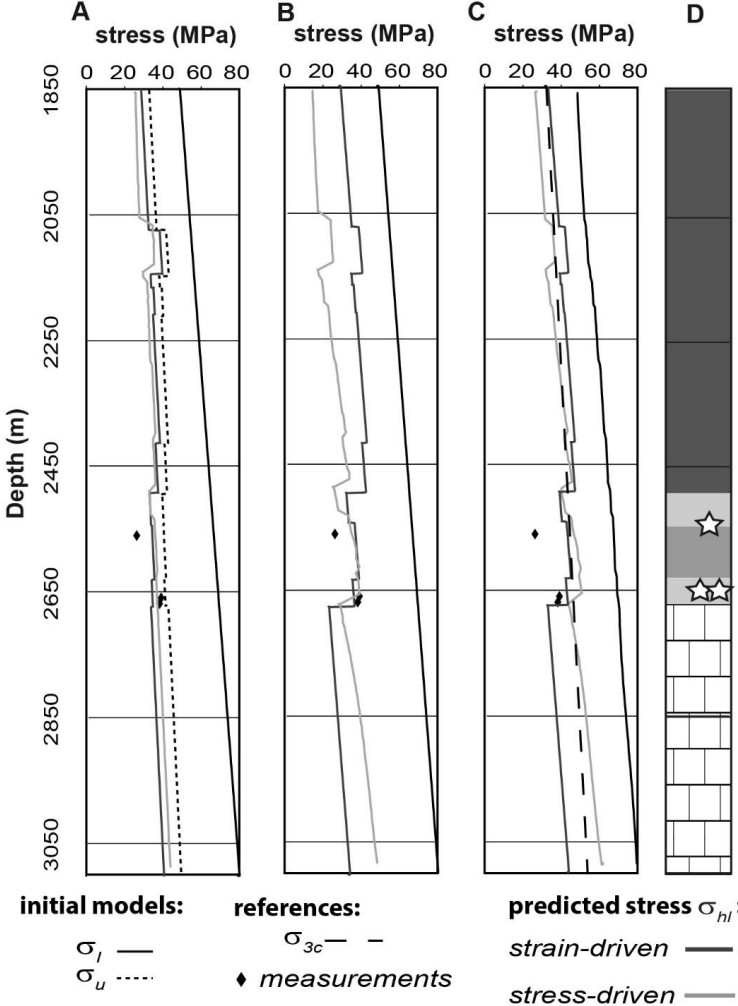

**Figure 6:** Comparison of the stress and strain-driven models. Stress predictions for the stress-driven and the strain-driven

models for (A) predictive strategies 1 and 5, based on the uniaxial stress and in situ stress measurements, (B) predictive

strategies 2 and 6, based on the lithostatic stress and in situ stress measurements, and (C) predictive strategies 4 and 8,



based on the lithostatic stress and critical model. Also shown are the initial uniaxial stresses $\sigma_u$ (A), the initial lithostatic stresses $\sigma_l$ (A-C) and the critical stresses $\sigma_{3c}$ (C). D: Simplified stratigraphic column of the studied area with the same legend as for Figure 4.