# Peer review of "Modeling of the in situ state of stress in elastic layered rock subject to stress and strain-driven tectonic forces"

_Solid Earth, 2016_

## Referee Comment (RC1) · Anonymous Referee #1 · 8 Dec 2016

Dear authors and editors,

The paper investigates different strategies to predict depth variation of stress within a sedimentary succession. This is an important issue as an increasing amount of stress models are applied to predict stresses in the upper crust, having only little data. The case study is based on the stratigraphy, log data and in-situ stress measurements from a well located in the Western Canada Sedimentary Basin. The paper is well written and the different strategies are well explained. Results are good portrayed and discussed. Therefore I suggest to accept the paper pending on minor revision.

In the following I will point out the major concerns followed by some smaller comments, which are always marked with the specific line or the figure number.

[Figure]

**General aspects**

The introduction chapter is in some parts is patchy and not straightforward. E.g. the paragraph from line 55 to 59 would fit much better before paragraph starting line 48. From my personal point of view, the appendix A (B) would fit well within the introduction or chapter 2. $S_V$, $S_H$ and $S_h$ are well known. However, the relative position to each other could be mentioned in the introduction. The used strategy (line 209-217) would be a good final paragraph for the introduction.

The general idea is to combine the initial stress model (lithostatic and uniaxial) with certain elastic material properties, a type of boundary conditions (stress- and strain driven) in comparison with a benchmark (in-situ stress data or criticality of potential faults). This is well explained in lines 209 to 213. However, the reader is often mislead that one of the model parts (e.g. uniaxial, strain driven or critical) are separate models. A small sketch showing the relative position of model parts/boundary conditions would help.

The case, that only three in-situ $S_h$ data are available within a short stratigraphic sequence is a drawback. However, the authors should not be judged for lacking available in-situ stress data.

**Minor issues with reference to specific lines or figure numbers**

line 25: both cited papers from Reiter el. al. and Reiter and Heidbach (2014) are not overview paper on crustal stresses in general, these papers are focused on crustal stresses in Alberta and Canada. May other paper would fit much better as a general introduction.

lines 108-120: introduction ?

lines 187-189: introduction ?

line 198: 'horizontal stress profile' ?

line 220: 'depth-dependent Poisson's Ratio'? or lithology dependent Poisson's Ratio?

line 245: The max. (regional) horizontal stress is 'negligible'. That could be an assumption, but the maximum horizontal stress as well the correction are introduced and mentioned by several equations in chapter 2. I think that could be mentioned in the introduction.

lines 254-256: discussion ?

lines 269-271: introduction ?

line 636: 'E' is introduced, but not explained except line 651.

line 642: 'A' and 'B' are introduced, but not explained. May avoid A and B by combining Eq. B 8 and B 9?

line 660: Atkinson

some references includes the month (line 805 and 810)

Table 1: I would write b) and c) in the head of the table

Figure 2: Usage of colours instead of several grey tones in the stress profiles would allow much better to distinguish the shown properties/stress profiles

Figure 4: It should be mentioned that Muskwa, Otter Park and Evie member are together the Horn River Formation, which is mentioned only once in line 291. 'The white stars represents the injection levels.' Injection level is may be a little bit ambiguous for the location of stress measurements.

Figure 5 and 6. Usage of colours instead of several grey tones in the stress profiles would allow much better to distinguish the shown stress profiles. The stratigraphic column (E and I respective D) should indicate the Keg river and Horn river formation, as they are discussed related to the figures. The stratigraphic column is not the same as shown in Fig 4. Sub-figure 5H is not explained.

---

## Author Comment (AC1) · 15 Dec 2016

**General aspects**

**RC1:** The introduction chapter is in some parts is patchy and not straightforward.
In several places we modified the structure of the introduction in order to improve the flow according to the comments.

**RC1:** the paragraph from line 55 to 59 would fit much better before paragraph starting line 48.
We moved these lines in the introduction.

**RC1:** From my personal point of view, the appendix A (B) would fit well within the introduction or chapter 2.
We agree that the appendix A and B could be part of the main text. However, although those calculations are important in order to understand the rational beyond stress calculations, we think that they are better in appendix because they are mainly reviews of calculation previously described and this make the manuscript more focused.

**RC1:** SV , SH and Sh are well known. However, the relative position to each other could be mentioned in the introduction.
We added a sentence in section 2.1 to specify Sh and SH.

**RC1:** The reader is often mislead that one of the model parts (e.g. uniaxial, strain driven or critical) are separate models. A small sketch showing the relative position of model parts/boundary conditions would help.
I am not sure we understand the comment and further details would be great. There is already a Table and a figure giving details on the modelling strategies.

**Minor issues**

**RC1:** line 25: both cited papers from Reiter el. al. and Reiter and Heidbach (2014) are not overview paper on crustal stresses in general, these papers are focused on crustal stresses in Alberta and Canada. May other paper would fit much better as a general introduction.
We added two references.

**RC1:** lines 108-120: introduction ?
The lines 116-120 have been moved in the introduction.

**RC1:** lines 187-189: introduction ?
We think that those line are unnecessary details for the introduction and fit better in this section.

**RC1:** line 198: 'horizontal stress profile' ?
We removed "horizontal"

**RC1:** line 220: 'depth-dependent Poisson's Ratio'? or lithology dependent Poisson's Ratio?
We replaced depth-dependent by lithology dependent.

**RC1:** line 245: The max. (regional) horizontal stress is 'negligible'. That could be an assumption, but the maximum horizontal stress as well the correction are introduced and mentioned by several equations in chapter 2. I think that could be mentioned in the introduction.
Several assumptions are made and described in the paper. However, we think that specifying this one in particular in the introduction will be unnecessary details for the introduction.

**RC1:** lines 254-256: discussion ?
These lines has been moved in the discussion

**RC1:** lines 269-271: introduction ?
These lines has been moved in the introduction

**RC1:** line 636: 'E' is introduced, but not explained except line 651.
We added a sentence to explain E

**RC1:** line 642: 'A' and 'B' are introduced, but not explained. May avoid A and B by combining Eq. B 8 and B 9?
We modified this part to make clear what is A and B

**RC1:** line 660: Atkinson
We modified the reference.

**RC1:** some references includes the month (line 805 and 810)
We modified the reference.

**RC1:** Table 1: I would write b) and c) in the head of the table
We modified the table.

**RC1:** Figure 2: Usage of colours instead of several grey tones in the stress profiles would allow much better to distinguish the shown properties/stress profiles
We added colours in the figure.

**RC1:** Figure 4: It should be mentioned that Muskwa, Otter Park and Evie member are together the Horn River Formation, which is mentioned only once in line 291. 'The white stars represents the injection levels.' Injection level is may be a little bit ambiguous for the location of stress measurements.
We added a sentence in the caption to specify that Muskwa, Otter Park and Evie member are together the Horn River Formation. We replaced injection levels by location of stress measurements.

**RC1:** Figure 5 and 6. Usage of colours instead of several grey tones in the stress profiles would allow much better to distinguish the shown stress profiles. The stratigraphic column (E and I respective D) should indicate the Keg river and Horn river formation, as they are discussed related to the figures. The stratigraphic column is not the same as shown in Fig 4. Sub-figure 5H is not explained.
We modified the figures, the captions and added colors.

---

## Referee Comment (RC2) · Anonymous Referee #2 · 24 Jan 2017

**Review on the manuscript entitled "Modeling of the in situ state of stress in elastic layered rock subject to stress and strain-driven tectonic forces" by Vincent Roche and Mirko van der Baan**

**MS No.: se-2016-141**

**General comments**

The manuscript deals with in situ stress estimation in layered elastic rock. This is an important issue since current knowledge and approaches are often not sufficient or precise enough or unsatisfactory. The study focusses on the influence of stress at horizontal boundaries where rocks of different elastic properties interact.

The state of stress regarded as being adequately predicted (defined as "reference stress") is assumed as either the state of stress from measurements or the state of stress corresponding to a critical state of stress from frictional theory. Starting with the assumption of two fundamental states of stress which are (1) stress under uniaxial strain and (2) lithostatic stress the layered rock is subject to horizontal stress or strain to attain the reference stress in the layered rock.

The manuscript is well written and well discussed. Nevertheless, I have a number of issues with this manuscript which I briefly state in the following. Further explanation is given under "specific comments".

1) It is claimed that the study is about stress prediction through modeling. In the manuscript stress prediction is being considered as kind of minimum difference between either stress magnitudes from measurements or a critical state of stress and the result of modeling. I wonder why the modeling is necessary. What is the benefit of modeling here, specifically in layered elastic rock? Considering the case study, the stress data along with the remarks in lines 330-333 already give useful information on the state of stress.

2) I would find it helpful to regroup the introduction and to rewrite certain passages or write them more precisely, in particular lines 73-78 which are in large part not understandable without knowing the following sections (see remarks in "specific comments").

3) I miss a description of the numerical models (see e.g. remarks to lines 141/142/244/245/263/630 and 191-193 and 565-569)

4) In general, the magnitude of the maximum horizontal stress is not well accessible. In the considered normal faulting case it corresponds to the intermediate principal stress. The approach is therefore difficult to apply in strike-slip or thrust faulting regimes as far as stress data from measurements are taken as "reference stress".

5) It is assumed that stress changes as well as strain changes are zero in the horizontal orientation perpendicular to the one considered. (I would find it an interesting topic to investigate if anywhere on Earth this assumption is valid). For further explanation see "specific comments". I would like to see something like a proof or explanation that this assumption does not affect the conclusions made.

6) In my understanding it seems that the stress-driven theoretical models are actually strain-driven as the same strain seems to be applied at each layer. In the strain-driven models the layers are said to be coupled whereas they seem to be uncoupled (see e.g. comments to Figures 1 and 2). In case of stress boundary conditions coupling at the interface between

layers does not exclude shear stress and different strain within the layers and different displacement at the boundaries, except at the interface.

**Specific comments**

Line 35/36: None of the mentioned stress measurement techniques allows measuring the full in situ stress; only part of it like minimum principal stress or orientation of maximum horizontal stress.

Line 58: "… by applying stress updates to initial stress models. This is a common methodology to predict stresses in the crust …"
Without knowing what is explained later on, I think this expression cannot be understood.

Line 62: "Or, this model can provide bounds for the minimum and maximum principal stresses"
The model can provide bounds for differential stress (but not of the magnitudes of $\sigma_1$ and $\sigma_3$) or for one principal stress if the other one is known.

Line 65: "Therefore the stress differentials cannot exceed their shear strength."
It is well possible that differential stress exceeds shear strength.

Line 68: "stress predictions from the elastic properties of rock", Line 69-70: "stress predictions based on elasticity are commonly used…"
How should that work?

Paragraph line 60-66 could be added to line 49-52.

Line 73: "1D stress prediction": Maybe it could be mentioned earlier that this article is essentially about the magnitude of the minimum horizontal stress and not about the full state of stress or about the orientation of stress.

Line 74: "the models": so far only theoretical models were introduced whereas here it is referred to numerical models. These have not been adequately introduced up to that point. Or just say "in numerical models" instead of "in the models".

Line 75: I think it cannot be understood at this point what "updating strategies" means (see comment to line 58).

Line 76: "Several models are used and combined". It cannot be understood at this point that the "models" that are combined mean very different things here such as: 1.) different assumptions on a general state of stress in the Earth, 2.) different calibration measures and 3.) different modeling approaches.

Lines 83/85/89/94/217/260/473-474: "…$\Delta_{hl}$ and $\Delta_{Hl}$ are the local tectonic stress corrections."
This implies that no other processes such as the ones mentioned in lines 55-58 are relevant, except tectonic stresses. However, in line 89: "The stress corrections reflect processes such as erosion, sedimentation, temperature change, tectonic activity."
To be consistent either only tectonic stress should be considered or all of the mentioned effects. In the letter case it is questionable whether all the different processes can be represented in one additive quantity, as these processes act very differently.

Lines 91/93/107/141: Until now there is no technique the magnitude of maximum horizontal stress can be measured with. There are some approaches for estimation of the magnitude of the maximum horizontal stress, however these involve assumptions and are far from being accurate.

Line 135: why "and Eq. (7)"? Eq. (8) and (9) refer only to Eq. (6).

Line 137: It should read $\Delta\sigma_{hl} = \Delta\varepsilon_{Hr} \; E\nu/(1-\nu^2)$ instead of $\Delta\sigma_{hl} = \Delta\varepsilon_{hr} \; E\nu/(1-\nu^2)$

Line 141/142/244/245/263/630: "The problem becomes more complex if the second lateral regional strain perturbation is not negligible." "… we postulate that only the minimum regional stress $\Delta\sigma_{hr}$ or strain $\Delta\varepsilon_{hr}$ perturbation exists and that the maximum regional stress $\Delta\sigma_{Hr}$ or strain $\Delta\varepsilon_{Hr}$ perturbation is negligible"
What would be the criterion to decide if it's negligible? The general assumption would be that it is not negligible. Is the approach also applicable if it is not negligible?
Is the modeling done as a 2D section or in 3D? If 2D: plane stress or plane stress? In case of plane stress: is there negligible strain change in the perpendicular direction in the model? In case of plane strain: is there negligible stress change in the perpendicular direction in the model? If 3D: what are the boundary conditions to ensure that there is neither stress change nor strain change in the perpendicular orientation in the model? In fact there must be subsidence or uplift if this condition is to be fulfilled …

Line 156/157: "since no stress transfer or interaction occurs between layers"
This should be only a theoretical case, as it is not realistic, isn't it?

Line 162/163: "if the layers are not coupled"
again a theoretical case. Or frictionless detachment …

Lines 191-193 and Figure 2B/C: numerical and analytical solutions are compared. It seems that in the analytical model boundary conditions at the base and top are not appropriately defined. How are they defined? Or how to explain the difference? See also comment to Figure 2C.

Line 215: The term "reference stresses" has not been defined yet. Maybe shift the bracket in the following line or the definition in lines 227/228 to line 215. Note also that the term "reference stress" in geosciences is generally considered as an idealized stress state in a static crust with no tectonic forces and is thus defined differently from what the term is used in this manuscript ("initial stress" as it is used in this manuscript is a reference stress in common understanding). Another word for "reference stress" as it is used in this manuscript could be "benchmark stress", "calibration stress" or something like that.

Line 221: "We assume a Biot pore-pressure coefficient equal to 1."
Line 291: "low porosity shale members"
A Biot coefficient of 1 is reasonable for unconsolidated high-porosity sands where external stress is taken up by pore pressure. In a tight shale I would expect the Biot coefficient to be much lower, perhaps 0.5-0.7.

Line 248: "… the regional stress perturbation is assumed constant across all layers". Constant stress in each layer implies different strain in each layer if Young's modulus is different.
What is the difference between Figures 1A/B and 1E/F? If in the coupled stress-driven case stress corrections are individual for each layer but then strain corrections are said to be the same for all layers then it is essentially the same as the strain-driven case. See also comment to Figure 1.

Line 260-264: conclusion would be: in situ stress measurements as reference stress can only be used for a normal faulting regime given the maximum horizontal stress, which cannot be adequately determined by measurements, would be needed for a strike-slip or thrust faulting regime?

Line 272-274: "For the strain-driven model, the local stress corrections $\sigma_{hl}$ are calculated analytically using the vertical variation of Poisson's ratio $\nu$ and Young's $E$ with Eq. (8). Then, the depth-dependent local stress $\sigma_{hl}$ is calculated with Eq. (1), the pore pressure $P_p$, and the appropriate initial stresses $\sigma_{hli}$." Do I understand correctly that the strain-driven case is not numerically modeled but analytically calculated according to Eq.(8)? Then there would be no coupling, contrary to the statement in line 265.

Line 338: "The lithostatic stress $\sigma_l$ increases linearly with depth"
Line 341: "the minimum horizontal critical stress $\sigma_{3c}$ increases linearly with depth,"
Which means same density for all layers. Figure 4B is just used for calculation of elastic parameters?

Line 453/454: "the in-situ stress measurements are substantially lower than the critical stresses"
This either means that the stress measurements are wrong or that wrong assumptions were made in calculating the critical stress. See also comment to Figure 5.

Lines 565-569: what is the numerical resolution? Maybe provide element size, element type and whether first or second order elements are being used. How many layers of elements were used to mesh the individual Horn River formations? Is the vertical change in Young's modulus and Poisson's ratio as depicted in Fig. 4 represented in the numerical model or is some smoothening applied?

Line 578: "the combined analysis of all eight stress predictions helps reveal the true uncertainty"
The word "true" makes this sentence untrue. If all of the models include the same systematic errors or wrong assumptions, all of the models can deviate from the true state of stress.

Line 599: (A.2) holds, if there is no poroelastic coupling acting, otherwise the term $\alpha\, P_p\, (1-2\nu)/(1-\nu)$ adds (Engelder and Fischer 1994).

Line 611, Eq. (A.3): why without $\alpha$ in contrast to Eq. (A.2)?

Line 619: Eq. (B.1) should hold only at the interface but not within the layers, if stress applied at the boundary is the same and Young's modulus is different in the layers.

Figure 1: in case of strain-driven extensional and coupled stress-driven extensional boundary conditions (figures at the top and bottom on the right side): it should read $\Delta\sigma_{hls} > \Delta\sigma_{hlc}$ instead of $\Delta\sigma_{hls} < \Delta\sigma_{hlc}$ if $\Delta\sigma_{hls}$ and $\Delta\sigma_{hlc}$ are absolute values. The same extensional strain applied to a stiff and a compliant material results in lower stress in the stiff material than in the compliant one and in a greater stress **change** in the stiff material than in the compliant one. This can be seen in Figures 5C and 5D in comparison with Fig. 4D, provided that the extensional strain is applied uniform over this depth section.

Figure 1: both, $\sigma_h$ and $\sigma_H$ change due to the boundary conditions. How does this fit with the assumption that stress change in the other direction is zero (Lines 141/142/244/245/263/630)?

Figure 1B and 1F: should be $\Delta\varepsilon_{hls} = \Delta\varepsilon_{hlc}$ instead of $\Delta\varepsilon_{Hls} = \Delta\varepsilon_{Hlc}$ ?

Figure 1: I see no difference between Figure 1A and E and between 1B and 1F. If strain is the same so will be stress (not in the layers but in cases 1A/E and 1B/F).

Figure 2C. If applied stress is -10 MPa then why should stress reduction be less than 10 MPa in the compliant layer and greater than 10 MPa in the stiff layer? Close to the interface I do expect this difference but not far above and below the interface. In other words, I expect what can be seen from the numerical solution in Figure 2B. Conclusion would be that coupling is not correctly represented in the analytical solution. This is probably due to the fact that not only stress but also strain is assumed to be the same at the boundary in case 2C, while it is not in case 2B. If stress boundary conditions are applied then horizontal displacement is not the same along the boundary due to the different stiffness of the layers. See discussion why there is no difference between Figures 1A/E and 1B/F.

The critical stress state in the figures is based on a uniform coefficient of friction. Maybe the clay-rich formations in the case study have a lower coefficient of friction than the carbonates?

Fig. 5: Why are cases 3 and 7 not being shown?

Fig. 5: The critical stress state $\sigma_{3c}$ exceeds all of the stress measurement data. In a normal faulting regime this cannot be the case as the vertical stress is the maximum principal stress and the stress measurements represent the minimum principal stress. This either means that the stress measurements are wrong or the coefficient of friction (and cohesion) assumed to calculate the critical stress state is wrong or the estimated vertical stress is wrong (wrong average density).

Figure 6: it seems that for the stress-driven cases applied stress change at the boundary is constant over the considered depth interval, at least in Figures 6B and 6C and that the stress boundary condition is adjusted for the depth of the measurements. Maybe it is more reasonable to assume an increase of stress change with depth starting with lower values at the top because stress becomes very small there and is over-critical. In contrast for the strain-driven cases stress changes not much with depth. Here as well a strain gradient could be more realistic when considering not only the depth of the measurements but the whole depth interval.

Figure 6: which criterion was used to adjust the state of stress to the reference stress? Minimum difference, least squares, eyeballed?

**Technical corrections**

Line 60: "… that the state of stress in the crust is close to the maximum strength that rocks can support…" This basically means a tensor is compared to a scalar value. Could be more precise.

Line 68: either "are" instead of "is" or "prediction" instead of "predictions"

Line 86: "an uniaxial strain model" instead of "a uniaxial model", see line 53.

Line 138/146/232/236/262/284: should read "normal faulting regime"

Line 153: should be "extension" not "shortening", compare Fig. 2

Line 170: should be $\Delta\sigma_{Hr}$ instead of $\Delta\sigma_{hr}$

Line 171: should be $\Delta\sigma_{hr}$ instead of $\Delta\sigma_{Hr}$

Line 172: exchange 1C/1D or $\sigma_h/\sigma_H$

Line 221: "either to the stress of the uniaxial strain model" would be more precise

Line 232: "normal faulting" instead of "normal fault"

Line 235: as the maximum horizontal stress cannot be measured, just estimated at best, one may erase some words "data point to calibrate  the minimum principal stress at a specific depth. This could be done with in-situ stress measurements "

Line 237: insert "coefficient of" before "friction"

Line 247: should be Eq.(2) instead of (1) and Eq.(3) instead of (2) ?

Line 264: maybe "minimum" instead of "minimal"

Line 278: "An in situ initial state of stress":  maybe erase "in situ"

Line 310: should be NE-SW ?

Line 311: does also a gross difference exist?

Line 322/326/327: MPa km$^{-1}$ , not MPa/Km$^{-1}$ or MPa.Km$^{-1}$

Line 338/342: maybe erase "slightly" because there is a substantial difference between the two

Line 353: why "even"? Should be always the case.

Line 375/406/413/428 etc.     I would find it helpful to insert "(Table 1)" after the strategy numbers a few more times

Line 384: "target" instead of "targets"

Line 442: should be "except", not "expect"?

Line 445: should be $\sigma_{3c} < \sigma_h$  and not $\sigma_{3c} > \sigma_h$ if "infra-critical" means not critical ($\sigma_l$ is the vertical stress, which is in a normal faulting regime greater than the horizontal stress, so the critical stress must be smaller or equal to the horizontal stress)

Line 446: should be $\sigma_{3c} > \sigma_h$ if "supra-critical" means more than critical

Line 455: maybe "requires" is better than "produces".

Line 462: "MPa" instead of "MPA"

Line 462: "formations" instead of "Formations".

Lines 484/489/492/596: "an uniaxial" instead of "a uniaxial"

Line 514: "fault" instead of "faults"

Line 525: maybe "elastic parameters" instead of "elasticity"

Line 554     thus are prone

Line 594: although it is about stress it should read "uniaxial strain model"

Line 596/597: "creates a uniaxial horizontal stress $\sigma_u$ due to the lateral extension of the medium that is impeded by non-deformable walls (i.e., no lateral strains)."
I understand the meaning but it sounds like a contradiction, maybe write differently

Line 614: "lower", not "higher"

Line 630: "…with zero stress and zero strain in the second horizontal direction…" It should read "stress change" instead of "stress"?

Line 640: "layer" not "layers"

Figure 2, diagrams on the right side: the shallowest depth should be 2200 instead of 2400 m

Line 847: must be extension not shortening

Line 848: Section 2.5 does not exist

Line 849: Sections 2.6 and 2.7 do not exist

Line 850: insert "m" after 2400

Line 850: maybe add density used to calculate the lithostatic stress

Figure 3: should be $\rho$ instead of $\nu$ below the box "lithostatic model"

behavior/behaviour:  sometimes with "u" (lines 306, 515, 557), sometimes without "u" (lines 28, 53, 57, 69, 81, 143, 156, 531, 616)

modelled/modeling, sometimes with two "l" (line 276), sometimes with one "l" (line 1,47, 77, 187, 191)

---

## Referee Comment (RC3) · Anonymous Referee #3 · 31 Jan 2017

General comments

Thank you for the opportunity to review the manuscript. The manuscript address an important topic of stress variations in layered formations, which are usually poorly known, and are not well understood. It is applicable to a variety of practical and fundamental problems in subsurface exploration and geomechanics. The authors' methodology is well explained and discussed. However, I have problems with the basic premise of the analysis, and with understanding the befits of, and the motivation for the chosen technique:

1) The title and the first sentence in the abstract suggest that the manuscript provides methods to predict stress variation for layered formations. The prediction strategies,

however, are based on looking at the difference between one of the two simple models (initial models) and either stress measurements or a more realistic prediction (from critical state-of-stress model). What this analysis does, in my view, is showing how far those two initial models are from reality. This could be of some value, too, but should be framed as an assessment of those models. In the current formulation, the following questions remain: Why would we need to model stress at the points where we have it measured? Why would we need to start with initial models, which do a poor job of assessing realistic stresses, if we have a better theory providing stress limits from the critical state-of-stress theory?

2) How does the comparison between initial models and the 'locally measured stresses' (lines 227-229) allows assessing the magnitude of 'tectonic effects'? Would not the difference be comprised of the tectonic effects PLUS the local stress perturbation due to stress/stress partitioning along layer boundaries?

3) Maximum horizontal stress cannot be measured directly, and therefore, cannot be used in the 'reference' model based on measurements. This fact is skipped over throughout the paper, including the introduction and the discussion.

Technical corrections: Lines 49-50 and lines 60-65 describing critical state-of-stress theory are repetitive, consider reorganizing

Line 68: should be '. . .predictions . . . are', not 'is'

Line 237: misplaced comma

Line 244-245: It's not clear whether this is a valid assumption

Line 308: should be 'constants ..are overestimated', not 'overestimate'

Line 871: should be kg/m3, not kg.m3

---

## Author Comment (AC2) · 27 Feb 2017

**Reply to reviewer 2**

Manuscript entitled "Modeling of the in situ state of stress in elastic layered rock subject to stress and strain-driven tectonic forces" by Vincent Roche and Mirko van der Baan

MS No.: se-2016-141

**General comments:**

**RC1:** It is claimed that the study is about stress prediction through modeling. In the manuscript stress prediction is being considered as kind of minimum difference between either stress magnitudes from measurements or a critical state of stress and the result of modeling. I wonder why the modeling is necessary. What is the benefit of modeling here, specifically in layered elastic rock? Considering the case study, the stress data along with the remarks in lines 330-333 already give useful information on the state of stress.

A few stress measurements and some derived information on stress gradients is indeed useful. However, data points are often sparse, and in many cases non-existent. Only if one believes that the horizontal stress is adequately described by a simple gradient would it be possible to extrapolate and interpolate measurement points. If not, modelling has to be performed in order to obtain insights into the vertical variation in stress both in between and beyond individual measurements, which is critical for many topics (failure tendency, slip tendency, fault growth, engineering problematic, fluid flow…).

**RC2:** I would find it helpful to regroup the introduction and to rewrite certain passages or write them more precisely, in particular lines 73-78 which are in large part not understandable without knowing the following sections (see remarks in "specific comments").

We modified the introduction in accordance to the specific comments.

**RC3:** I miss a description of the numerical models (see e.g. remarks to lines 141/142/244/245/263/630 and 191-193 and 565-569).

We added information on the numerical modelling (see specific comments)

**RC4:** In general, the magnitude of the maximum horizontal stress is not well accessible. In the considered normal faulting case it corresponds to the intermediate principal stress. The approach is therefore difficult to apply in strike-slip or thrust faulting regimes as far as stress data from measurements are taken as "reference stress".

Indeed, stress assessment become more complex for strike-slip regime and thrust fault regime. We modified the manuscript to take into account this comment and according to the specific comments (l.35, l.49, l.551-558).

**RC5:** It is assumed that stress changes as well as strain changes are zero in the horizontal orientation perpendicular to the one considered. (I would find it an interesting topic to investigate if anywhere on Earth this assumption is valid). For further explanation see "specific comments". I would like to see something like a proof or explanation that this assumption does not affect the conclusions made.

We agree this is an important assumption in our modelling and the specific results for the case history. However since one of the objectives of the paper is to show the variability in stress

predictions given the implemented predictive strategy, adding additional stress predictions based on exploring the contribution of the stress/strains in the second horizontal direction seems counterproductive. It will indeed increase the spread and variability in predictions but at the expense of adding an entire new layer of parameters. Since the conclusion is that stress predictions are inherently uncertain, this specific assumption does not invalidate the conclusion.

**RC6:** In my understanding it seems that the stress-driven theoretical models are actually strain-driven as the same strain seems to be applied at each layer. In the strain-driven models the layers are said to be coupled whereas they seem to be uncoupled (see e.g. comments to Figures 1 and 2).
The layers in the strain-driven models have the same displacements; coupling or no-coupling is not involved in this case. In this model, strain is imposed as boundary condition. In the stress-driven model, the strain is resulting from the stress boundary condition and depend on the local characteristics (layer thicknesses and stiffnesses).
In case of stress boundary conditions coupling at the interface between layers does not exclude shear stress and different strain within the layers and different displacement at the boundaries, except at the interface.
Indeed for the fig 1 and for the analytic formulation, we assume that the strain is constant within a layer and between two surrounding layers for the stress-driven model. The figure 1 is a theoretical case that aims to illustrate the model and hence the accent is put on the main effect. For the analytic formulation, the problem become far more complex if strain variations have to be taken into account within a layer. In the numerical modelling there is no assumption on the local strain.

**Specific comments:**

Line 35/36: None of the mentioned stress measurement techniques allows measuring the full in situ stress; only part of it like minimum principal stress or orientation of maximum horizontal stress.
We modified the introduction to provide this information (l.35-39).

Line 58: "… by applying stress updates to initial stress models. This is a common methodology to predict stresses in the crust …" Without knowing what is explained later on, I think this expression cannot be understood.
We modified this sentence (l.54)

Line 62: "Or, this model can provide bounds for the minimum and maximum principal stresses" The model can provide bounds for differential stress (but not of the magnitudes of $\sigma_1$ and $\sigma_3$) or for one principal stress if the other one is known.
 We modified the sentence in accordance to the comment

Line 65: "Therefore the stress differentials cannot exceed their shear strength." It is well possible that differential stress exceeds shear strength.
We modified the sentence (l.70).

Line 68: "stress predictions from the elastic properties of rock", Line 69-70: "stress predictions based on elasticity are commonly used…" How should that work?
 We do not understand this comment, but we considerably modified the section.

Paragraph line 60-66 could be added to line 49-52.

We modified this paragraph in the introduction according to the comment.

Line 73: "1D stress prediction": Maybe it could be mentioned earlier that this article is essentially about the magnitude of the minimum horizontal stress and not about the full state of stress or about the orientation of stress.
This is now better stated in the manuscript (l.49, l.78)

 Line 74: "the models": so far only theoretical models were introduced whereas here it is referred to numerical models. These have not been adequately introduced up to that point. Or just say "in numerical models" instead of "in the models".
We modified the paragraph and deleted "in the models".

Line 75: I think it cannot be understood at this point what "updating strategies" means (see comment to line 58).
We modified the paragraph

Line 76: "Several models are used and combined". It cannot be understood at this point that the "models" that are combined mean very different things here such as: 1.) different assumptions on a general state of stress in the Earth, 2.) different calibration measures and 3.) different modeling approaches.
We modified the paragraph to take into account the comment (l.78-94).

Lines 83/85/89/94/217/260/473-474: "…$\sigma_{hl}$ and $\sigma_{Hl}$ are the local tectonic stress corrections."  This implies that no other processes such as the ones mentioned in lines 55-58 are relevant, except tectonic stresses. However, in line 89: "The stress corrections reflect processes such as erosion, sedimentation, temperature change, tectonic activity." To be consistent either only tectonic stress should be considered or all of the mentioned effects. In the letter case it is questionable whether all the different processes can be represented in one additive quantity, as these processes act very differently.
In the paper, we focus on the tectonic effect. Other processes are cited for reference. We remove this line in the section in order to clarify this point (l.105).

Lines 91/93/107/141: Until now there is no technique the magnitude of maximum horizontal stress can be measured with. There are some approaches for estimation of the magnitude of the maximum horizontal stress, however these involve assumptions and are far from being accurate.
This is now clearly indicated in several part of the manuscript (l.35, l.49, l.551-558).

Line 135: why "and Eq. (7)"? Eq. (8) and (9) refer only to Eq. (6).
We deleted "equ. (7) ".

Line 137: It should read $\sigma_{hl} = \sigma_{Hr} E\sigma/(1-\nu^2)$ instead of $\sigma_{hl} = \sigma_{hr} E\sigma/(1-\nu^2)$
Yes! It was a misspelling.

Line 141/142/244/245/263/630: "The problem becomes more complex if the second lateral regional strain perturbation is not negligible." "… we postulate that only the minimum regional stress $\sigma_{hr}$ or strain $\varepsilon_{hr}$ perturbation exists and that the maximum regional stress $\sigma_{Hr}$ or strain $\varepsilon_{Hr}$ perturbation is negligible". What would be the criterion to decide if it's negligible? The general assumption would be that it is not negligible. Is the approach also applicable if it is not negligible?

The importance of the assumption is dependent on both the Poisson's ratios as well as the actual correction in the second horizontal direction. As a rule of thumb, if the predicted corrections are small then the assumption is also valid. If larger corrections are predicted then, it may be worth investigating the sensitivity of the results with respect to this assumption. This assumption is an important one, but all other assumptions investigated in the 8 predictive strategies are also important.

Is the modeling done as a 2D section or in 3D? If 2D: plane stress or plane stress? In case of plane stress: is there negligible strain change in the perpendicular direction in the model? In case of plane strain: is there negligible stress change in the perpendicular direction in the model? If 3D: what are the boundary conditions to ensure that there is neither stress change nor strain change in the perpendicular orientation in the model? In fact there must be subsidence or uplift if this condition is to be fulfilled.

Details of the numerical modelling approach are now provided in more detail in section 3.4 (l.306). The modeling is performed in 3D, only stress boundaries are applied, and the model is free to deform in all directions. The model is set in equilibrium as an in-situ state of stress is applied within the model. Then, the stress condition are modified at the boundary of the model (change in minimum principal stress). As a consequence, there are local strain modifications in all directions, including in the direction perpendicular to the model due to Poisson ratio. The local strain and the change in vertical stress is minor (there is no tectonic effect in this direction).

Line 156/157: "since no stress transfer or interaction occurs between layers" This should be only a theoretical case, as it is not realistic, isn't it?

It is a theoretical case, but this should be the case assuming such a model dominates in nature, although there should be stress continuity at all internal interfaces. To go further, the layers are actually not totally independent in the strain driven model because the regional strain is computed using an average value of the local strain in each layer. Changing a layer could therefore affect the regional strain and hence the stress in the other layers.

Line 162/163: "if the layers are not coupled" again a theoretical case. Or frictionless detachment …

This is a theoretical case used to understand the mechanism, but indeed similar behaviour is expected in the case of layering rock with layers separated with a frictionless detachment (this is discussed in section 6.3).

Lines 191-193 and Figure 2B/C: numerical and analytical solutions are compared. It seems that in the analytical model boundary conditions at the base and top are not appropriately defined. How are they defined? Or how to explain the difference? See also comment to Figure 2C.

This is indeed a problem of boundary condition for those theoretical cases. Those model are here to illustrate the mechanism. In the analytic solution for simplification in the formulation we considered that the strain is constant within a layer. That strain corresponds to the strain at the interface because we focus here in the effect of the layering. In fact for such a case, strain is likely to change far from the interface. This behaviour is reproduced by the numerical model, resulting in the stress variation within a layer. However in a real sedimentary pile, stress transfer should occurs at both the lower and upper interfaces, so that the strain and the stress will become more constant inside the layer, which is what we see in fig 6, and alike when we consider constant strain.

Line 215: The term "reference stresses" has not been defined yet. Maybe shift the bracket in the following line or the definition in lines 227/228 to line 215. Note also that the term "reference

stress"in geosciences is generally considered as an idealized stress state in a static crust with no tectonic forces and is thus defined differently from what the term is used in this manuscript ("initial stress" as it is used in this manuscript is a reference stress in common understanding). Another word for "reference stress" as it is used in this manuscript could be "benchmark stress", "calibration stress" or something like that.

We replaced reference by calibration in all manuscript.

Line 221: "We assume a Biot pore-pressure coefficient equal to 1." Line 291: "low porosity shale members" A Biot coefficient of 1 is reasonable for unconsolidated high-porosity sands where external stress is taken up by pore pressure. In a tight shale I would expect the Biot coefficient to be much lower, perhaps 0.5-0.7.

We agree that a more complex pore pressure profile could be used. Influence of pore pressure in stress prediction is analysed in another paper (Roche and Van der Baan, 2015). But much more has to be done to understand the coupling between pore pressure and stress state.

Line 248: "… the regional stress perturbation is assumed constant across all layers". Constant stress in each layer implies different strain in each layer if Young's modulus is different. What is the difference between Figures 1A/B and 1E/F? If in the coupled stress-driven case stress corrections are individual for each layer but then strain corrections are said to be the same for all layers then it is essentially the same as the strain-driven case. See also comment to Figure 1.

We are distinguishing between the boundary (or regional) stresses and the local stresses (as measured in the center of our numerical models). In the figure 1A/B, a regional strain, independent of the local characteristic of the model is applied, resulting in local stress change in each layer depending on the properties of the layers. In the figure 1E/F, a regional stress independent of the local characteristic of the model is applied, resulting in local stress and strain change in each layer depending on the properties of the layers.

Line 260-264: conclusion would be: in situ stress measurements as reference stress can only be used for a normal faulting regime given the maximum horizontal stress, which cannot be adequately determined by measurements, would be needed for a strike-slip or thrust faulting regime?

We added a paragraph in the discussion concerning this point (l.554-558).

Line 272-274: "For the strain-driven model, the local stress corrections $\Delta$hl are calculated analytically using the vertical variation of Poisson's ratio $\nu$ and Young's E with Eq. (8). Then, the depth-dependent local stress $\sigma$hl is calculated with Eq. (1), the pore pressure Pp, and the appropriate initial stresses $\sigma$hli." Do I understand correctly that the strain-driven case is not numerically modeled but analytically calculated according to Eq.(8)?

Yes, the strain driven model is calculated analytically and there is not coupling in this case.

Then there would be no coupling, contrary to the statement in line 265.

In this line we refer to the stress-driven model and not the strain driven model. For the stress driven model, we used a numerical modelling because the analytic solution for a complex sedimentary pile is complex.

Line 338: "The lithostatic stress $\sigma$l increases linearly with depth". Line 341: "the minimum horizontal critical stress $\sigma$3c increases linearly with depth," Which means same density for all layers. Figure 4B is just used for calculation of elastic parameters?

Indeed variation in density is used for calculation of elastic parameters, but it is not integrated in the stress prediction for simplicity. The rational is that variation in density has a stronger impact in

elastic parameters than in the vertical stress and in the pore pressure. For the vertical stress, a quick calculation indicates a difference lower than 0.3 MPa when integration the variation in density. Likewise, using an average density also allow us to avoid variation in stress gradients due to the calibration stress and therefore highlight better variations in gradient due to the tectonic effect.

Line 453/454: "the in-situ stress measurements are substantially lower than the critical stresses" This either means that the stress measurements are wrong or that wrong assumptions were made in calculating the critical stress. See also comment to Figure 5.
See comments figure 5.

Lines 565-569: what is the numerical resolution? Maybe provide element size, element type and whether first or second order elements are being used. How many layers of elements were used to mesh the individual Horn River formations? Is the vertical change in Young's modulus and Poisson's ratio as depicted in Fig. 4 represented in the numerical model or is some smoothening applied?
We provided additional information in section 3.4 (l.304)

Line 578: "the combined analysis of all eight stress predictions helps reveal the true uncertainty" The word "true" makes this sentence untrue. If all of the models include the same systematic errors or wrong assumptions, all of the models can deviate from the true state of stress.
We deleted "true" in the sentence.

Line 599: (A.2) holds, if there is no poroelastic coupling acting, otherwise the term ⬚ Pp (1-2⬚)/(1-⬚) adds (Engelder and Fischer 1994).
We added this sentence in the manuscript (l.644).

Line 611, Eq. (A.3): why without $\alpha$ in contrast to Eq. (A.2)?
We added the $\alpha$.

Line 619: Eq. (B.1) should hold only at the interface but not within the layers, if stress applied at the boundary is the same and Young's modulus is different in the layers.
True, in a real case, the strain is likely to change far from the interface. Such a variation in strain is adding more complexity. See previous comment for more details.

Figure 1: in case of strain-driven extensional and coupled stress-driven extensional boundary conditions (figures at the top and bottom on the right side): it should read ⬚⬚hls > ⬚⬚hlc instead of ⬚⬚hls < ⬚⬚hlc if ⬚⬚hls and ⬚⬚hlc are absolute values. The same extensional strain applied to a stiff and a compliant material results in lower stress in the stiff material than in the compliant one and in a greater stress change in the stiff material than in the compliant one. This can be seen in Figures 5C and 5D in comparison with Fig. 4D, provided that the extensional strain is applied uniform over this depth section.
Yes it is true for absolute values. Here, the values are negative in the case of extensional regime. So a lower value means a higher absolute value. I think it is better to not use absolute values has we are presenting both extension and compression in the figure.

Figure 1: both, ⬚h and ⬚H change due to the boundary conditions. How does this fit with the assumption that stress change in the other direction is zero (Lines 141/142/244/245/263/630)?

In those lines, we refer to the regional perturbation that is assume to be negligible in the direction of the maximum horizontal stress. Local variation in sh and sH can rise from a regional perturbation in one direction. In the paper we focus on the variation in sh.

Figure 1B and 1F: should be ⬚⬚hls = ⬚⬚hlc instead of ⬚⬚Hls = ⬚⬚Hlc ?
Yes we modified the figure.

Figure 1: I see no difference between Figure 1A and E and between 1B and 1F. If strain is the same so will be stress (not in the layers but in cases 1A/E and 1B/F).
This figure is theoretical, the left boundaries are different but the local stresses trend the same. The figure shows the effect on the wall.

Figure 2C. If applied stress is -10 MPa then why should stress reduction be less than 10 MPa in the compliant layer and greater than 10 MPa in the stiff layer? Close to the interface I do expect this difference but not far above and below the interface. In other words, I expect what can be seen from the numerical solution in Figure 2B. Conclusion would be that coupling is not correctly represented in the analytical solution. This is probably due to the fact that not only stress but also strain is assumed to be the same at the boundary in case 2C, while it is not in case 2B. If stress boundary conditions are applied then horizontal displacement is not the same along the boundary due to the different stiffness of the layers. See discussion why there is no difference between Figures 1A/E and 1B/F.
This point is discussed in a previous remark.

The critical stress state in the figures is based on a uniform coefficient of friction. Maybe the clay-rich formations in the case study have a lower coefficient of friction than the carbonates?
We added a sentence to discuss that point (l.544)

Fig. 5: Why are cases 3 and 7 not being shown?
We added a sentence in the caption to clarify this point.

Fig. 5: The critical stress state ⬚3c exceeds all of the stress measurement data. In a normal faulting regime this cannot be the case as the vertical stress is the maximum principal stress and the stress measurements represent the minimum principal stress. This either means that the stress measurements are wrong or the coefficient of friction (and cohesion) assumed to calculate the critical stress state is wrong or the estimated vertical stress is wrong (wrong average density).
Yes we were puzzled by this too. Since we have no information on the cohesion of faults or the coefficient of friction we used standard estimates. The average density came from well logs which covered only a few lithologies. Since the objective of the paper is to show the inherent uncertainty in stress predictions this does not invalidate our conclusions. We added sentences in the section 6.2 in order to notice the role of changing friction and cohesion and uncertainty in in-situ stress measurement.

Figure 6: it seems that for the stress-driven cases applied stress change at the boundary is constant over the considered depth interval, at least in Figures 6B and 6C and that the stress boundary condition is adjusted for the depth of the measurements. Maybe it is more reasonable to assume an increase of stress change with depth starting with lower values at the top because stress becomes very small there and is over-critical. In contrast for the strain-driven cases stress changes not much

with depth. Here as well a strain gradient could be more realistic when considering not only the depth of the measurements but the whole depth interval.

Yes, whether it is the strain or the stress driven model, the boundary condition are constant all along the sedimentary pile. I agree that we can maybe think about a more complex model with increasing stress or strain with depth, but this is adding an extra complexity that is beyond the scope of this paper.

Figure 6: which criterion was used to adjust the state of stress to the reference stress? Minimum difference, least squares, eyeballed?

I do not understand clearly this comment. The boundary condition is adjusted using the average difference between the calibration and the initial stress.

**Technical corrections**

We take into account all technical corrections, except the following point:

Line 455: maybe "requires" is better than "produces". We used " induces".

Lines 484/489/492/596: "an uniaxial" instead of "a uniaxial"  English is not our native language but stress is on 'u' in this case so we strongly belief 'a uni-axial' is correct like in 'a uniform' and 'a university'.

Line 596/597: "creates a uniaxial horizontal stress ▯u due to the lateral extension of the medium that is impeded by non-deformable walls (i.e., no lateral strains)." I understand the meaning but it sounds like a contradiction, maybe write differently. Any suggestions? We're not sure how to formulate this differently.

---

## Author Comment (AC4) · 27 Feb 2017

**Reply to reviewer 3**

Manuscript entitled "Modeling of the in situ state of stress in elastic layered rock subject to stress and strain-driven tectonic forces" by Vincent Roche and Mirko van der Baan

MS No.: se-2016-141

**General aspects**

**RC1:** The title and the first sentence in the abstract suggest that the manuscript provides methods to predict stress variation for layered formations. The prediction strategies, however, are based on looking at the difference between one of the two simple models (initial models) and either stress measurements or a more realistic prediction (from critical state-of-stress model). What this analysis does, in my view, is showing how far those two initial models are from reality. This could be of some value, too, but should be framed as an assessment of those models. In the current formulation, the following questions remain:

Why would we need to model stress at the points where we have it measured?
If one had absolute faith in the measurement then there's no need to model these points. But these are not absolute measurements but inferred values. Their estimation depends strongly on chosen parameters and even personal preferences. Hence the modelling can be required to give additional confidence in their validity. But mostly modelling is used to obtain insights both in between and beyond individual measurements in surrounding layer.

Why would we need to start with initial models, which do a poor job of assessing realistic stresses, if we have a better theory providing stress limits from the critical state-of-stress theory?
In various cases the initial models give an accurate representation of the in situ stress field and the bounds depend strongly on the material constants (e.g., cohesion and internal angle of friction) which are not necessarily well known. If the bounds are narrow then they are certainly useful in numerous applications (predictions of slip tendency, caprock integrity, fault growth etc) but generally the bounds are quite wide. Likewise, to our knowledge, it does not allow directly to take into account complex heterogeneous system.

**RC2:** How does the comparison between initial models and the 'locally measured stresses' (lines 227-229) allows assessing the magnitude of 'tectonic effects'? Would not the difference be comprised of the tectonic effects PLUS the local stress perturbation due to stress/stress partitioning along layer boundaries?
It is right that the difference between initial models and the 'locally measured stresses' may not be only the magnitude of the tectonic effect and may contain several effects. We assume that one effect dominates.

**RC3:** Maximum horizontal stress cannot be measured directly, and therefore, cannot be used in the 'reference' model based on measurements. This fact is skipped over throughout the paper, including the introduction and the discussion.
We modified the manuscript to take into account this comment (l.35-39, section 6.2).